# The E3 ubiquitin ligase TRIM23 regulates adipocyte differentiation via stabilization of the adipogenic activator PPARγ

**Masashi Watanabe[1], Hidehisa Takahashi[1], Yasushi Saeki[2], Takashi Ozaki[1], Shihori Itoh[1], Masanobu Suzuki[1], Wataru Mizushima[1], Keiji Tanaka[2], Shigetsugu Hatakeyama[1]\***

[1]Department of Biochemistry, Hokkaido University Graduate School of Medicine, Sapporo, Japan; [2]Laboratory of Protein Metabolism, Tokyo Metropolitan Institute of Medical Science, Tokyo, Japan

**Abstract** Adipocyte differentiation is a strictly controlled process regulated by a series of transcriptional activators. Adipogenic signals activate early adipogenic activators and facilitate the transient formation of early enhanceosomes at target genes. These enhancer regions are subsequently inherited by late enhanceosomes. PPARγ is one of the late adipogenic activators and is known as a master regulator of adipogenesis. However, the factors that regulate PPARγ expression remain to be elucidated. Here, we show that a novel ubiquitin E3 ligase, tripartite motif protein 23 (TRIM23), stabilizes PPARγ protein and mediates atypical polyubiquitin conjugation. TRIM23 knockdown caused a marked decrease in PPARγ protein abundance during preadipocyte differentiation, resulting in a severe defect in late adipogenic differentiation, whereas it did not affect the formation of early enhanceosomes. Our results suggest that TRIM23 plays a critical role in the switching from early to late adipogenic enhanceosomes by stabilizing PPARγ protein possibly via atypical polyubiquitin conjugation.

**\*For correspondence:** hatas@ med.hokudai.ac.jp

**Competing interests:** The authors declare that no competing interests exist.

## Introduction

Adipose tissue is a major storehouse for excess energy and has recently been shown to be an endocrine tissue critical for the generation of hormones and cytokines involved in energy metabolism (*Galic et al., 2010*). Abnormalities in adipocyte differentiation or functions may result in the development of metabolic syndrome by inducing insulin resistance, and these abnormalities are likely to be induced in part by inappropriate regulation of gene expression required for adipocyte differentiation or functions.

The conversion of preadipocytes to mature adipocytes is a strictly controlled process regulated by a series of transcriptional activators. Both in vivo and in vitro studies have shown that peroxisome proliferator-activated receptor γ (PPARγ) and three CCAAT/enhancer family members (α, β, and δ) are essential regulators of adipogenesis (*Tontonoz et al., 1994b*; *Yeh et al., 1995*; *Barak et al., 1999*; *Imai et al., 2004*; *Sugii et al., 2009*). The adipogenic program is generally composed of two steps. The first step is initiated by adipogenic stimuli that induce early adipogenic activators including C/EBPβ, C/EBPδ and Krüppel-like factors (KLFs) and lead to the formation of early enhanceosomes (*Mori et al., 2005*; *Oishi et al., 2005*). In the second step, these early enhanceosomes induce late adipogenic activators including PPARγ and C/EBPα that lead to terminal differentiation by inducing genes necessary for the characteristics of mature adipocytes (*Rosen et al., 2002*; *Lefterova et al., 2008*).

Recent studies using high-throughput sequencing have shown that transcriptional activators do not function individually but rather cooperate with other activators and form hotspots at specific genomic

**eLife digest** The world is facing a global epidemic of obesity, which also increases the risk for diabetes and heart disease. Obesity is caused when excess fat is stored in fat cells, and overweight individuals have larger fat cells compared to healthy weight people. Therefore understanding how fat cells are created in the body can provide new ways to combat obesity.

Fat cells, also known as adipocytes, arise from precursor cells via a process called adipogenesis. This requires the activity of proteins called transcription factors that bind to DNA and switch on the expression of genes. PPARγ is an important transcription factor that drives the expression of the genes that are needed to convert a precursor cell to a mature adipocyte.

For adipogenesis to proceed, cells have to maintain the appropriate levels of PPARγ. If the amount of PPARγ bound to DNA is too low, then it is unable to activate gene expression. However, the mechanisms by which cells maintain the correct levels of PPARγ activity remain poorly understood. Watanabe et al. analyzed this process in mouse cells and identified a protein called TRIM23 that is produced in precursor cells. Cells in which the levels of TRIM23 were artificially lowered failed to mature into fat cells; this suggests that this protein is necessary for adipogenesis. Furthermore, in the absence of TRIM23, the amount of PPARγ that occupied regions of DNA was also markedly reduced. A direct consequence of this was a decline in the expression of several genes that are required for the later steps in the adipogenesis process.

Watanabe et al. next analyzed the mechanism through which TRIM23 had an effect on the levels of PPARγ. It is known from previous work that TRIM23 belongs to a family of enzymes that attach a small molecular tag called ubiquitin onto other proteins. This ubiquitin tag typically marks these proteins for rapid destruction by a large molecular machine called the proteasome. Watanabe et al. found that TRIM23 also modified PPARγ with ubiquitin, but that it did so in an unusual manner that instead prevented the proteasome from recognizing PPARγ and destroying it. As such, TRIM23 stabilizes the levels of PPARγ in cells.

By providing new insights into how adipogenesis is regulated, these findings suggest that TRIM23 may be a potential therapeutic target in the treatment of diabetes and disorders related to obesity.

regions (*Moorman et al., 2006*; *Chen et al., 2008*; *He et al., 2011*; *Siersbaek et al., 2011*; *Boergesen et al., 2012*; *Gerstein et al., 2012*). In addition, the existence of super-enhancers has been recently reported (*Loven et al., 2013*; *Whyte et al., 2013*). Super-enhancers are large genomic domains occupied by master transcriptional activators and mediators, which induce expression of genes that define cell identity, and are especially characterized by a large amount of localization of Mediator subunit 1 (MED1). During the early phase of adipogenic differentiation, hotspots are central constituents in super-enhancers and have been suggested to cooperate with super-enhancers to drive the differentiation process (*Siersbaek et al., 2014*). A previous study showed that C/EBPβ, C/EBPδ, CREB1 and PPARγ function as candidate master transcriptional activators in adipose tissue (*Hnisz et al., 2013*). Of those master transcriptional activators, PPARγ plays a central role in adipogenesis, whereas other factors cannot induce adipocyte differentiation in the absence of PPARγ (*Farmer, 2006*; *Rosen and MacDougald, 2006*).

PPARγ has two isoforms, PPARγ1 and PPARγ2. These two isoforms are generated from alternate promoter usage and splicing, and PPARγ2 contains 30 additional amino acids at the amino-terminus (*Zhu et al., 1995*; *Fajas et al., 1997*; *Tontonoz and Spiegelman, 2008*). PPARγ1 is ubiquitously expressed and PPARγ2 is strictly expressed in adipose tissues, while both isoforms are strongly induced during adipocyte differentiation. Although ectopic expression of either of the PPARγ isoforms can induce adipocyte differentiation, PPARγ2 is thought to play a more central role in this process (*Mueller et al., 2002*; *Zhang et al., 2004*). It has been reported that PPARγ expression and activity are regulated at different levels such as transcription, protein degradation and post-translational modification (*van Beekum et al., 2009*; *Eeckhoute et al., 2012*; *Ahmadian et al., 2013*; *Lee and Ge, 2014*).

Tripartite motif-containing (TRIM) proteins (also known as E3 ubiquitin–protein ligases) are characterized by the presence of a RING finger, one or two zinc-binding motifs called B-boxes, and an associated coiled-coil region (RBCC), and there are currently known to be 77 TRIM proteins in

humans. TRIM family proteins are involved in a broad range of biological processes, and their alterations result in diverse pathological conditions (*Meroni and Diez-Roux, 2005*; *Ozato et al., 2008*; *Hatakeyama, 2011*). TRIM23 is a member of the TRIM family and possesses carboxy-terminal ARF (ADP ribosylation factor) domains. A recent study has shown that TRIM23 mediates atypical lysine 27 (K27)-linked polyubiquitin conjugation to NEMO, which plays an important role in the NFκB pathway, and this conjugation is essential for TLR3- and RIG-I/MDA5-mediated antiviral innate and inflammatory responses (*Arimoto et al., 2010*).

In this study, we identified TRIM23 as a novel factor that regulates PPARγ protein stability, possibly via atypical ubiquitin conjugation to PPARγ. TRIM23 knockdown caused reduction of PPARγ protein levels, which were restored by treatment with a proteasome inhibitor, leading to a severe defect in adipogenic differentiation. TRIM23 is dispensable for the early adipogenic program but is indispensable for the late adipogenic program. By controlling PPARγ abundance, TRIM23 functions as a regulator for a critical link between early and late enhanceosomes.

## Results

### TRIM23 is expressed in mouse adipose tissue during adipogenesis

To determine whether TRIM23 is involved in adipocyte differentiation, we first examined the expression of TRIM23 during differentiation of 3T3-L1 cells and in mouse adipose tissue. Real-time PCR analysis and immunoblot analysis revealed that TRIM23 was expressed in preadipocytes and that mRNA and protein levels slightly increased during adipogenesis (*Figure 1A,B*). Consistent with these results, TRIM23 was found to be predominantly expressed in the preadipocyte-containing stromal vascular fraction (SVF) at a level as high as that in the mature adipocyte fraction (*Figure 1C*). We also measured *Trim23* expression in a model of diet-induced obesity, and we found no significant difference between the expression levels in mice receiving the high-fat diet and those receiving the control chow (*Figure 1C*). We next examined the subcellular localization of TRIM23. We fractionated 3T3-L1 cells into nuclear extracts and cytoplasmic S100 fraction, and we found that a large amount of TRIM23 was distributed in the cytoplasm (*Figure 1D*).

### TRIM23 is required for adipogenesis

We next examined whether TRIM23 expression was necessary for adipocyte differentiation. Two different shRNAs (shTRIM23a and shTRIM23b) and a non-targeting control shRNA (shControl) were introduced into 3T3-L1 preadipocytes using retroviral vectors. One shRNA (shTRIM23a) efficiently depleted and the other (shTRIM23b) weakly depleted TRIM23 expression levels in 3T3-L1 preadipocytes relative to shControl (*Figure 2A*). These cells were stimulated to differentiate, and the ability to undergo differentiation to mature adipocytes was evaluated by determination of lipid accumulation using Oil Red O staining, direct measurement of intracellular triglyceride contents and determination of relative mRNA levels of adipocyte-specific genes including *Fabp4*, *Cidec*, *Klf15*, *Adipoq*, and *Retn*. Remarkably, TRIM23 knockdown significantly decreased lipid accumulation and markedly impaired the induction of adipocyte-specific genes (*Figure 2B–D* and *Figure 2—figure supplement 1*). To test whether this regulatory network is relevant to human adipocytes, we introduced siRNA into human primary visceral preadipocytes and differentiated them to mature adipocytes. Consistent with the results for mouse 3T3-L1 cells, TRIM23 knockdown significantly decreased lipid accumulation (*Figure 2B* and *Figure 2—figure supplement 2*). These findings indicate that TRIM23 is required for efficient conversion of preadipocytes to mature adipocytes. It has been shown that most adipocyte-specific genes are PPARγ target genes (*Lefterova et al., 2008*; *Nielsen et al., 2008*). We next examined the occupancy of PPARγ at a well-described PPARγ target gene, *Fabp4*, during adipocyte differentiation. PPARγ forms heterodimers with RXRα, which specifically bind to PPAR response elements (PPREs). It has been reported that PPARγ is recruited to two PPREs located 5500 bp upstream from the transcriptional start site (TSS) of the *Fabp4* gene, and this recruitment mediates activation of the *Fabp4* gene (*Figure 2E*) (*Tontonoz et al., 1994a*; *Nielsen et al., 2006*). We found that TRIM23 depletion decreased the occupancy of PPARγ on the enhancer PPRE at day 4 using chromatin immunoprecipitation (ChIP)-qPCR analysis (*Figure 2F*). It has been shown that some subunits of Mediator complex are necessary for the adipogenic process. MED14 is required for full activation of PPARγ-mediated transcription and adipocyte differentiation in vitro, and MED23 is required for the early transcriptional events during adipogenesis (*Wang et al., 2009*;

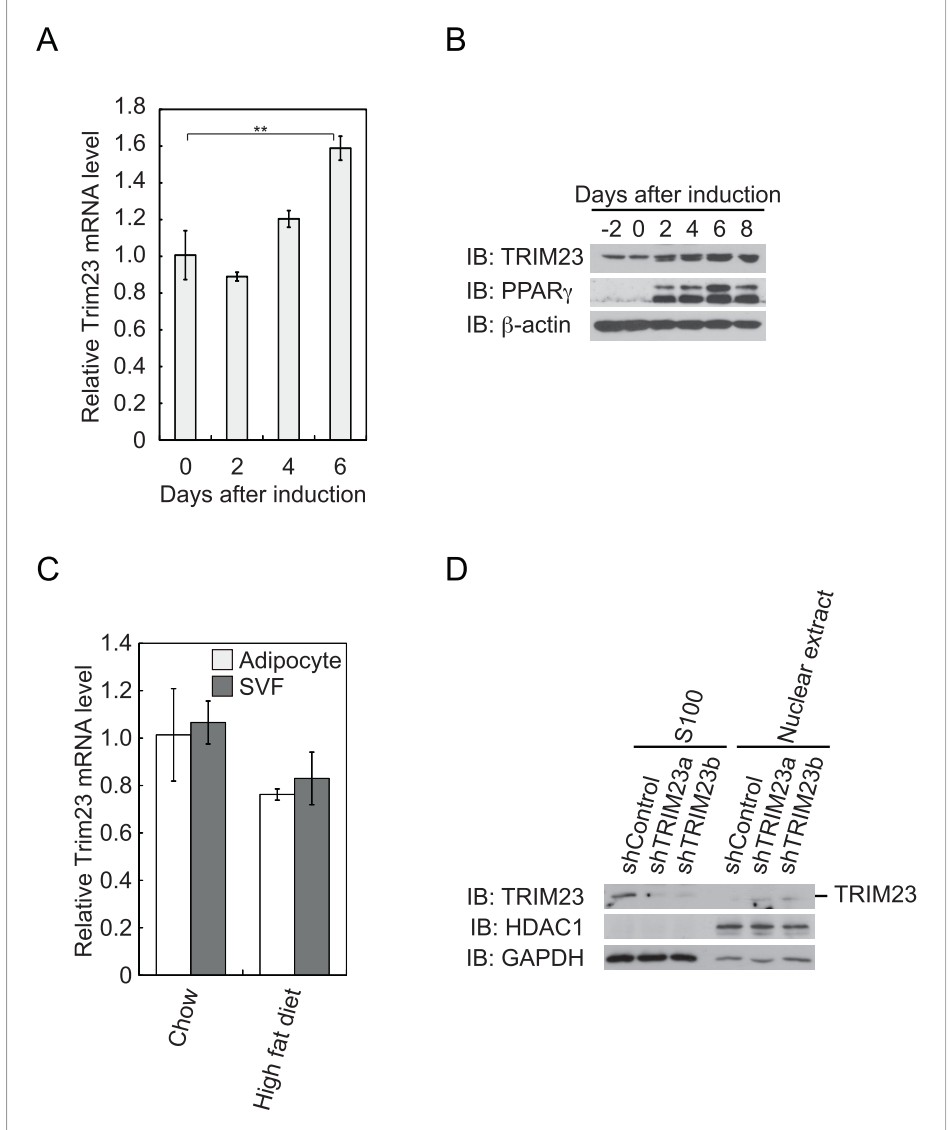

**Figure 1**. TRIM23 is expressed during adipocyte differentiation. (**A**) Real-time PCR analysis of mRNA expression of *Trim23* during 3T3-L1 differentiation. Total RNA was isolated from 3T3-L1 cells on the indicated days. *Trim23* mRNA was normalized to that of *Gtf2b*. (**B**) Immunoblot analysis of TRIM23 and PPARγ protein during 3T3-L1 cell differentiation. (**C**) Real-time PCR analysis of mRNA expression of *Trim23* in mouse adipose tissue. *Trim23* mRNA was normalized to that of *Gtf2b*. (**D**) Subcellular localization of TRIM23. Nuclear extracts and cytoplasmic S100 fraction were prepared from 3T3-L1 cells, and immunoblot analysis of TRIM23, HDAC1, and GAPDH was performed.

*Grontved et al., 2010*). MED1 is also required for adipogenesis in vitro (*Ge et al., 2002*). We tested the occupancy of MED1, one of the Mediator subunits, and found that TRIM23 depletion reduced the occupancy of MED1 at the *Fabp4* gene at day 4 (*Figure 2G*). We also found reduced occupancy of Pol II at the *Fabp4* gene in TRIM23 knockdown cells (*Figure 2H*). These findings suggest that the adipogenic defect by TRIM23 knockdown is caused by reduced PPARγ recruitment and subsequent reduced transcriptional activation on the target genes.

## TRIM23 is required for induction of late adipogenic activators but not for induction of early adipogenic activators during adipogenesis

We showed that TRIM23 knockdown reduces PPARγ recruitment to the enhancer and subsequent Pol II recruitment to the promoter at the target genes. To elucidate mechanisms of decreased

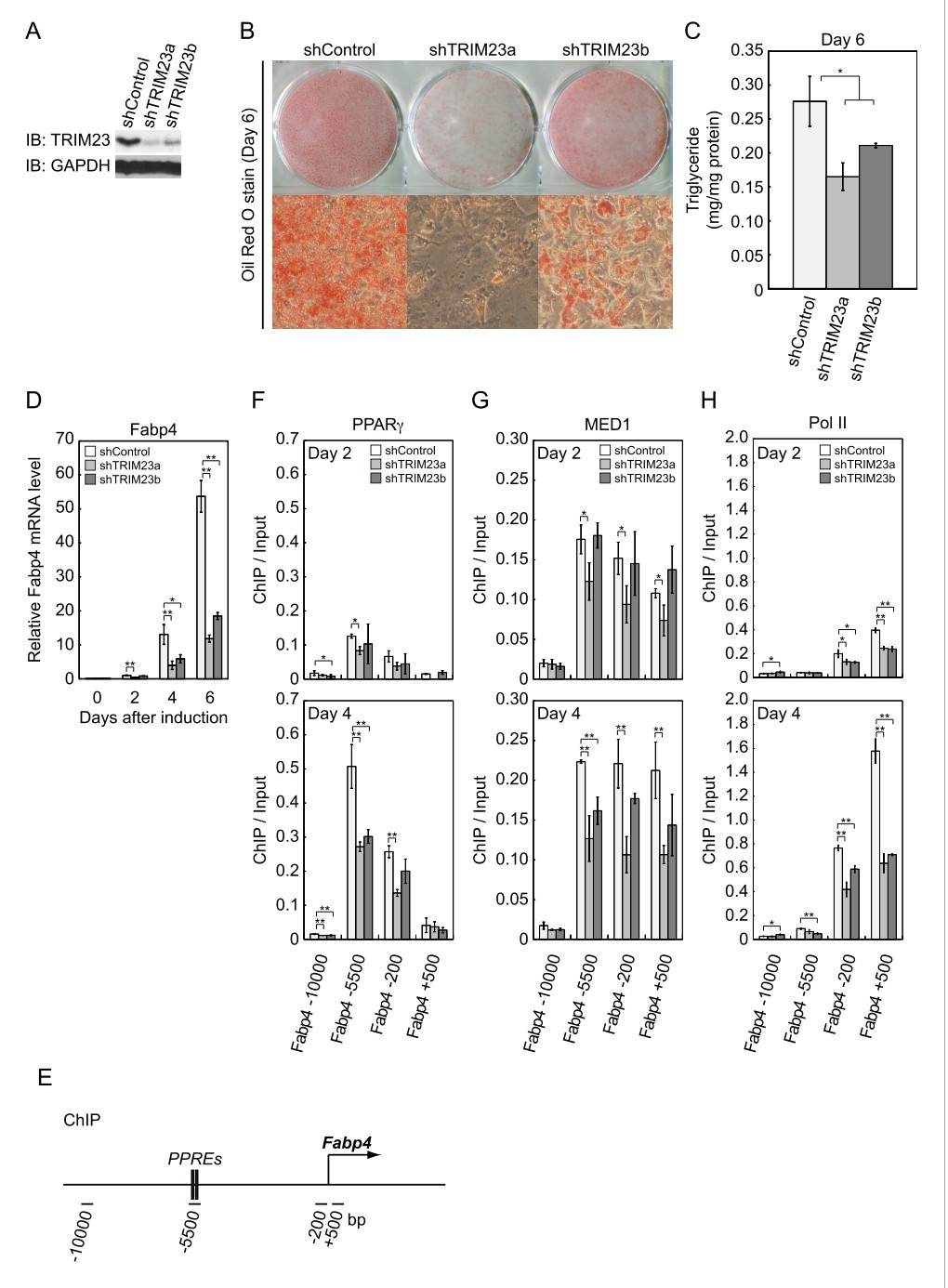

**Figure 2**. TRIM23 is required for 3T3-L1 adipocyte differentiation. (**A**) Immunoblot analysis of TRIM23 knockdown before induction of adipogenesis is shown. (**B**) Cells were stained with Oil Red O to visualize the accumulation of lipid droplets at day 6. (**C**) The amounts of intracellular triacylglyceride (TG) were quantified at day 6. (**D**) RNA levels of *Gtf2b* and *Fabp4* were determined by real-time PCR at days 0, 2, 4 and 6. Expression level of each gene was normalized to that of the *Gtf2b*. (**E**) Schematic representation of the *Fabp4* gene. The locations of the sequences amplified in the ChIP are shown at the bottom in base pairs relative to the *Fabp4* transcriptional start site. (**F**, **G** and **H**) ChIP analysis of PPARγ (**F**), MED1 (**G**), and Pol II (**H**) on the *Fabp4* gene during adipocyte differentiation. Ct values of each ChIP were normalized to that of input. All data represent means ± s.d. from three independent experiments. The p values for the indicated comparisons were determined by Student's *t* test (*, p < 0.05; **, p < 0.01).

The following figure supplements are available for figure 2:

*Figure 2. continued on next page*

*Figure 2. Continued*

**Figure supplement 1**. Quantitative analysis of *Trim23, Cidec, Klf15, Adipoq* and *Retn* mRNA during 3T3-L1 differentiation.

**Figure supplement 2**. TRIM23 is required for human visceral preadipocyte differentiation.

PPARγ-mediated gene activation in TRIM23 knockdown 3T3-L1 cells, we investigated mRNA levels of several adipogenic transcription factors during adipogenesis. Real-time PCR analysis revealed that induction of early transcriptional factors, *Cebpb* and *Cebpd*, was largely unaffected during adipocyte differentiation and that the expression of late adipogenic activators, *Pparg* and *Cebpa*, was equally induced at day 2 but was significantly reduced from day 4 by TRIM23 depletion (*Figure 3*). These findings suggest that an adipogenic defect by TRIM23 knockdown is caused by the decreased expression of late adipogenic activators.

## TRIM23 knockdown does not affect the occupancy of C/EBPβ and C/EBPδ at the PPARγ2 promoter during adipogenesis

It has been reported that hormonal treatment of 3T3-L1 cells acutely induces the expression of C/EBPβ and C/EBPδ within a few hours after stimulation. These early transcriptional activators have also been shown to be recruited to their target sites including the *Pparg* locus (*Steger et al., 2010*; *Siersbaek et al., 2011*). C/EBPβ marks a subset of early transcription factor hotspots and forms early enhanceosomes with other early adipogenic transcription factors (*Siersbaek et al., 2011*). These early enhanceosomes facilitate epigenetic modification and chromatin remodeling. To determine whether TRIM23 affects PPARγ induction, we investigated the expression levels of *Cebpb* and *Cebpd* and subsequent recruitment of C/EBPβ and C/EBPδ to the *Pparg* locus at the early phase of adipogenesis. As shown in *Figure 4A*, there were little differences in *Cebpb* and *Cebpd* mRNA levels between TRIM23 knockdown and controls. We next examined the possibility that TRIM23 affected the occupancy of C/EBPβ and C/EBPδ at the *Pparg* promoter during adipogenesis using ChIP-qPCR analysis (*Figure 4B,C*). As previously reported, C/EBPβ and C/EBPδ were enriched to their target sites, including *Pparg, Iqck*, and *Cav2* loci (*Siersbaek et al., 2011*), but TRIM23 depletion had no significant effect on this enrichment. These results suggest that TRIM23 functions at the downstream stage after recruitment of early adipogenic activators.

During differentiation, growth-arrested 3T3-L1 preadipocytes synchronously re-enter the cell cycle and undergo several rounds of clonal expansion, called mitotic clonal expansion (MCE) (*Tang et al., 2003b*). MCE is required for expression of adipogenic proteins and progression of terminal differentiation. C/EBPβ has been shown to play a pivotal role in MCE (*Tang et al., 2003a*). To evaluate MCE in TRIM23-knockdown cells, we counted cell numbers during adipocyte differentiation (*Figure 4D*). However, there was no difference between TRIM23-knockdown cells and the corresponding control. These findings also support the idea that TRIM23 does not affect the activities of C/EBPβ. Taken together, we concluded that PPARγ2 expression by TRIM23 is not regulated through induction and occupancy of C/EBPβ and C/EBPδ at the *Pparg2* promoter.

## Depletion of TRIM23 does not affect epigenetic marks and chromatin opening but decreases the occupancy of PPARγ itself and Pol II at the *Pparg* promoter

Activators such as C/EBPβ and C/EBPδ promote the recruitment of general transcriptional machinery composed of general transcription factors (GTFs) and RNA Pol II to form the pre-initiation complex (PIC) (*Lemon and Tjian, 2000*; *Thomas and Chiang, 2006*). To achieve this goal, activators recruit GTFs and Pol II directly or indirectly through binding to many cofactors. The transcription cofactors can be largely classified into two groups. The first group is composed of covalent histone-modifying enzymes and ATP-driven nucleosome remodelers to reorganize the chromatin architecture (*Belotserkovskaya and Berger, 1999*; *Jaenisch and Bird, 2003*; *Margueron et al., 2005*). These factors create an open chromatin environment suitable for transcription. The second group consists of general cofactors such as the Mediator complex and the SAGA complex (*Timmers and Tora, 2005*).

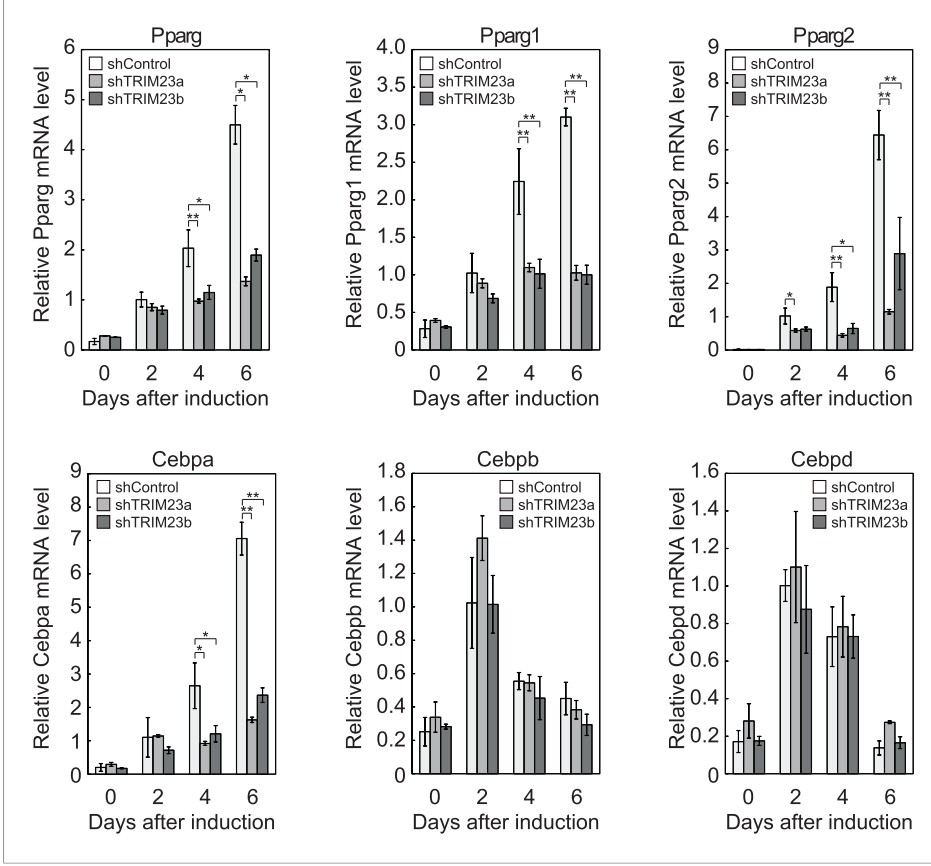

**Figure 3**. TRIM23 is required for induction of *Pparg1*, *Pparg2* and *Cebpa* but not for induction of *Cebpb* and *Cebpd* during adipogenesis. RNA levels of *Pparg*, *Cebpa*, *Cebpb* and *Cebpd* were determined by real-time PCR at days 0, 2, 4 and 6. Expression level of each gene was normalized to that of the *Gtf2b*. The data represent means ± s.d. from three independent experiments. The p values for the indicated comparisons were determined using Student's *t*-test (*, p < 0.05; **, p < 0.01).

The Mediator complex is an evolutionarily conserved coregulatory complex that typically works in communication between activators and general transcription machinery (*Malik and Roeder, 2010*). It has been shown that some subunits of the Mediator complex are necessary for the adipogenic process (*Ge et al., 2002*; *Wang et al., 2009*; *Grontved et al., 2010*).

To clarify the steps of *Pparg* transcription at which TRIM23 acts after activator recruitment, we observed epigenetic changes at the *Pparg* promoters during adipogenesis by ChIP-qPCR analysis. Trimethylation of histone H3 lysine 4 (H3K4me3), acetylation of histone H3 lysine 27 (H3K27ac), and monomethylation of histone H4 lysine 20 (H4K20me1) correlate with gene activation and increase at the *Pparg* gene early in the differentiation process, whereas dimethylation of histone H3 lysine 9 (H3K9me2) correlates with gene repression and decreases during adipogenesis (*Fujiki et al., 2009*; *Tie et al., 2009*; *Wakabayashi et al., 2009*; *Mikkelsen et al., 2010*; *Wang et al., 2013*). H3K4me3 and H3K27ac levels at *Pparg* gene increased during adipocyte differentiation, and there was little change caused by TRIM23 depletion (*Figure 5A,B*). There were no differences in H4K20me1 and H3K9me2 levels at day 2 between TRIM23-knockdown 3T3-L1 cells and corresponding control cells (*Figure 5—figure supplement 1*). These results indicate that TRIM23 does not affect epigenetic marks at the *Pparg* promoter. It has also been reported that the *Pparg* promoter is reorganized by ATP-driven nucleosome remodelers and that C/EBPβ binding is required for its chromatin opening. Therefore, we monitored chromatin opening using formaldehyde-assisted isolation of regulatory elements (FAIRE) analysis (*Giresi et al., 2007*). We performed FAIRE analysis at days 0 and 2, and we observed increased chromatin opening around the C/EBPβ binding sites on the *Pparg* promoter at

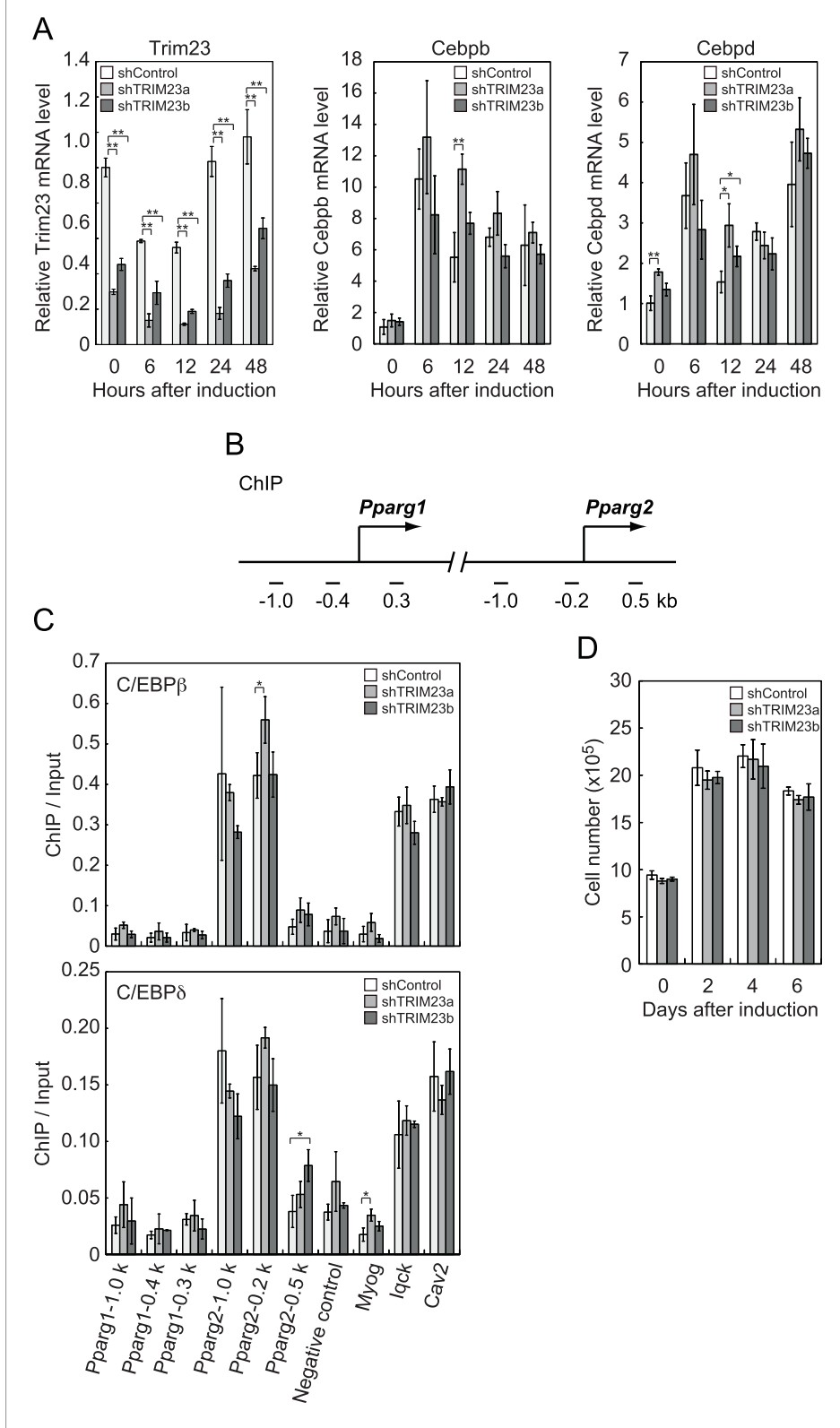

**Figure 4**. TRIM23 knockdown does not affect the induction and occupancy of C/EBPβ and C/EBPδ at the *Pparg* promoter during adipogenesis. (**A**) RNA levels of *Trim23*, *Cebpb* and *Cebpd* were determined by real-time PCR at 0, 6, 12, 24 and 48 hr. Expression level of each gene was normalized to that of *Gtf2b*. (**B**) Schematic representation of the *Pparg* promoters. The locations of the sequences amplified in the ChIP are shown at the bottom in base pairs

*Figure 4. Continued*

relative to the *Pparg1* and *Pparg2* transcriptional start sites. (**C**) Occupancy of C/EBPβ and C/EBPδ on a subset of target genes at 4 hr. Ct values of each ChIP were normalized to that of input. (**D**) Cell proliferation analysis of 3T3-L1 cells during adipocyte differentiation. 3T3-L1 cells were counted at each time point. The data represent means ± s.d. from three independent experiments. The p values for the indicated comparisons were determined by Student's *t* test (\*, $p < 0.05$; \*\*, $p < 0.01$).

day 2, while there was little difference caused by TRIM23 knockdown (*Figure 5C*). Since PPARγ by itself enhances the expression of PPARγ2 by binding to the *PPRE* at the *Pparg* promoter, we performed ChIP-qPCR analysis with a PPARγ antibody and found that the occupancy of PPARγ at the *Pparg* promoter was significantly decreased at day 4 by TRIM23 depletion (*Figure 5D*). In accordance with the data for PPARγ, the occupancy of Pol II at the *Pparg* promoter was equally increased at day 2 but was decreased at day 4, although there was little change in the occupancy of MED1 at day 2 and 4 by TRIM23 depletion (*Figure 5E,F*). Taken together, knockdown of TRIM23 did not affect formation of early enhanceosomes, epigenetic changes, chromatin remodeling, or recruitment of Pol II at the *Pparg* promoter during early adipocyte differentiation, but knockdown of TRIM23 reduced the formation of late enhanceosomes and the recruitment of Pol II during late adipogenic differentiation.

## TRIM23 does not affect PPARγ transcriptional activity in HEK293T cells

It has been reported that a positive feedback loop between C/EBPα and PPARγ expression supports the maintenance of the differentiated stage (*Rosen et al., 2002*). Both C/EBPα and PPARγ binding sites exist at the *Pparg* promoter, and PPARγ binding sites also exist at the *Cebpa* locus (*Farmer, 2006*; *Nielsen et al., 2008*). PPARγ activates its own expression via direct targeting and/or via C/EBPα binding to the *Pparg* promoter. If transcriptional activity of PPARγ is directly enhanced by TRIM23, knockdown of TRIM23 should lead to decreased PPARγ expression. This hypothesis is important to clarify a discontinuity in PPARγ expression between early and late adipocyte differentiation. To examine whether TRIM23 mediates PPARγ expression via the transcriptional activity of PPARγ, we measured the activity using a dual luciferase system. TRIM23 showed slight suppression of PPARγ2-mediated transcriptional activity but did not show a dose-dependent relationship (*Figure 6A*). It has been reported that the transcriptional activity of PPARγ was modulated by SUMO-1 modification and that some TRIM proteins have SUMO E3 activities (*van Beekum et al., 2009*; *Chu and Yang, 2011*). We therefore investigated whether TRIM23 mediates SUMOylation of PPARγ2. Although PPARγ2 was definitely SUMOylated in cells, TRIM23 did not affect this modification (*Figure 6B*). These findings suggest that changes in the transcriptional activity of PPARγ were not critical for impairment of PPARγ induction in TRIM23-knockdown cells.

## TRIM23 knockdown affects the stability of PPARγ1 and PPARγ2 proteins

To elucidate the mechanism of reduction of PPARγ expression by TRIM23 knockdown during late adipogenesis, we examined the expression of adipogenic activators during adipocyte differentiation at the protein level. Immunoblot analysis revealed that TRIM23 knockdown did not affect induction of early adipogenic activators but that TRIM23 knockdown impaired induction of late adipogenic activators (*Figure 7A*). Protein levels of PPARγ1 and PPARγ2 were already decreased at day 2 by TRIM23 knockdown (*Figure 7B*), whereas mRNA level of PPARγ1 was hardly reduced and that of PPARγ2 was moderately reduced in TRIM23-knockdown cells (*Figure 3*). Since the amount of PPARγ protein was more greatly reduced than the amount of *Pparg* mRNA in TRIM23-knockdown cells, we speculated that the stability of PPARγ protein was impaired by TRIM23 knockdown. It has been reported that the stability of PPARγ protein is regulated by the ubiquitin proteasome systems (*Floyd and Stephens, 2002*). To determine whether reduction of PPARγ protein in TRIM23-knockdown cells depends on proteasome activity, we observed the levels of PPARγ protein in the presence and absence of a proteasome inhibitor, MG132. 3T3-L1 cells were differentiated for 48 hr and subsequently treated with MG132. Immunoblot analysis showed that administration of MG132 blocked the reduction of both PPARγ1 and PPARγ2 protein levels in TRIM23-knockdown cells (*Figure 7C*). We also examined protein levels of PPARγ in preadipocytes and found that administration of MG132 blocked the reduction of PPARγ in TRIM23-knockdown cells

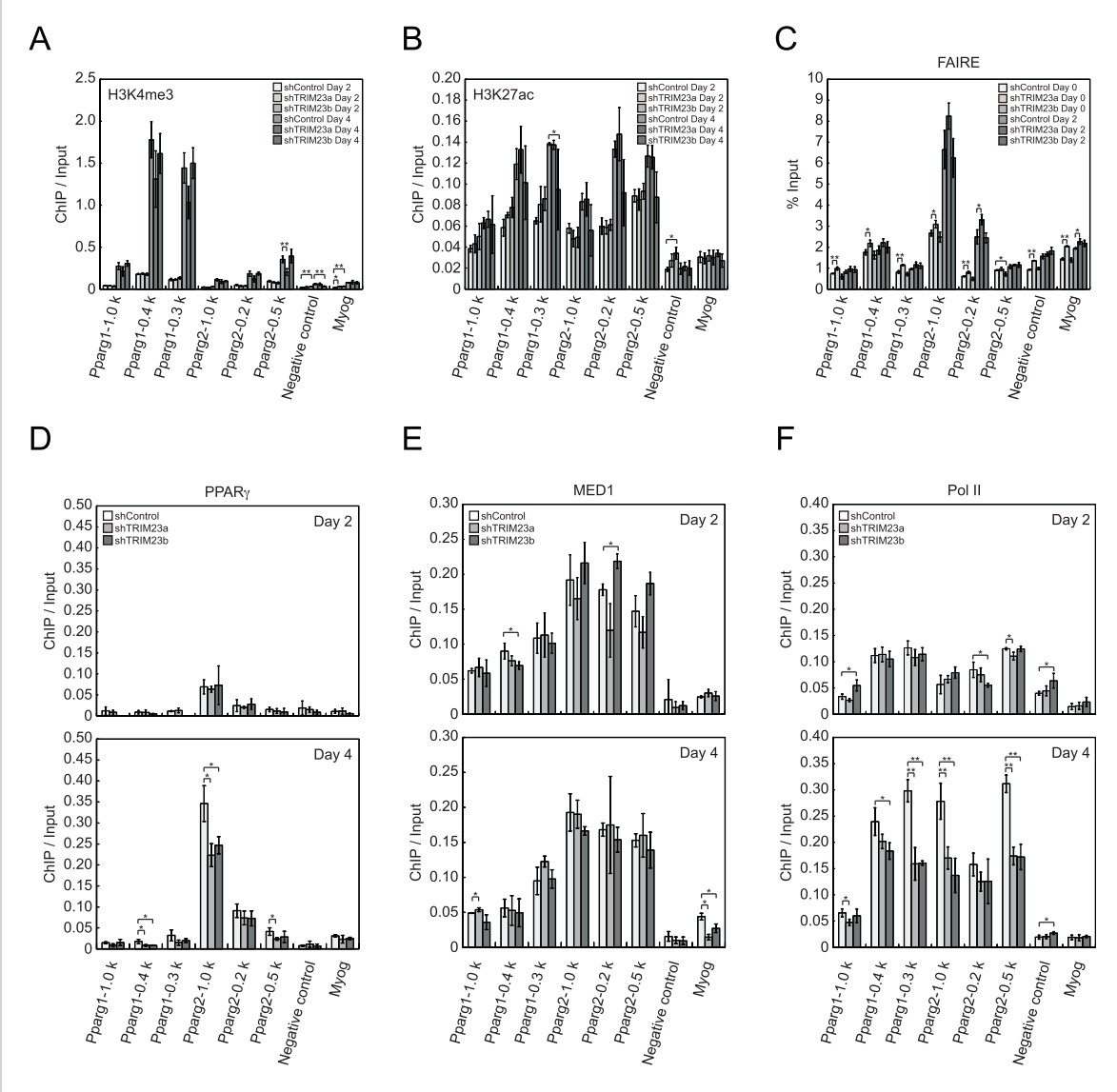

**Figure 5**. TRIM23 knockdown does not affect the epigenetic marks for activated genes and chromatin opening but decreases the occupancy of Pol II at the *Pparg* promoters. (**A** and **B**) ChIP analysis of H3K4me3 (**A**) and H3K27ac (**B**) at the *Pparg* promoters during adipocyte differentiation. Ct values of each ChIP were normalized to that of input. (**C**) Formaldehyde-assisted isolation of regulatory elements (FAIRE) analysis during adipocyte differentiation. The enrichment of fragmented genomic DNA in the FAIRE samples was analyzed and normalized to that of input. (**D**, **E** and **F**) ChIP analysis of PPARγ (**D**), MED1 (**E**), and Pol II (**F**) at the *Pparg* promoter during adipocyte differentiation. Ct values of each ChIP were normalized to that of input. All data represent means ± s.d. from three independent experiments. The p values for the indicated comparisons were determined by Student's *t* test (*, p < 0.05; **, p < 0.01).

The following figure supplement is available for figure 5:

**Figure supplement 1**. Depletion of TRIM23 does not affect H3K9me2 and H4K20me1 marks at the *Pparg* promoter.

---

(*Figure 7—figure supplement 1*). These findings indicated that the presence of TRIM23 is sufficient to inhibit degradation of PPARγ not only in a differentiating state but also in an undifferentiated state. Next, we examined the protein stability of PPARγ1 and PPARγ2. TRIM23-knockdown 3T3-L1 cells and the corresponding control cells were differentiated by the differentiation cocktail for 48 hr, and were subsequently treated with MG132 for 6 hr and with cycloheximide (CHX) for the indicated times. TRIM23 knockdown promoted the degradation of PPARγ1 and PPARγ2 (*Figure 7D,E*). These results suggest that TRIM23 stabilized PPARγ1 and PPARγ2 proteins during adipocyte differentiation.

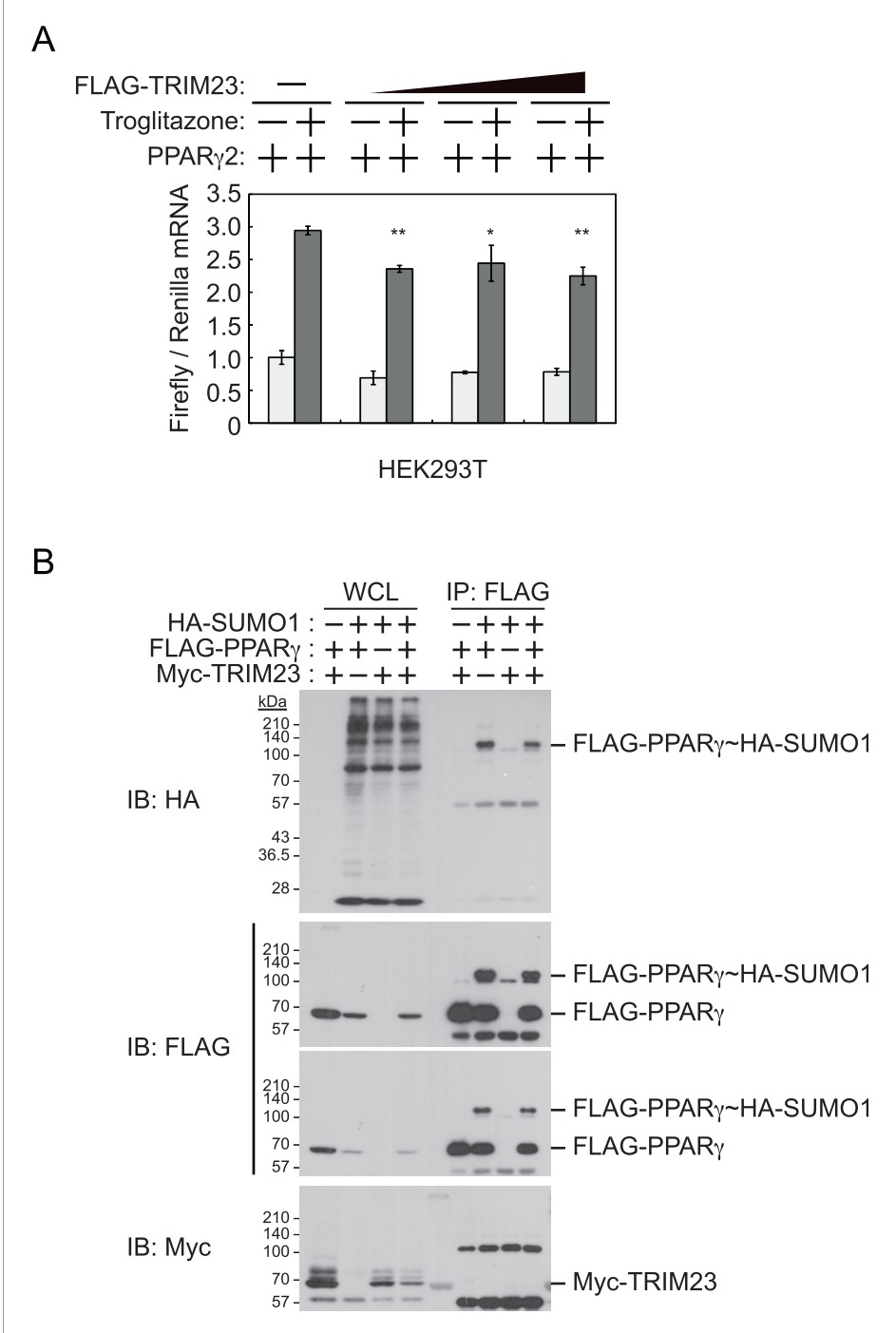

**Figure 6**. TRIM23 does not affect PPARγ2 transcriptional activity. (**A**) TRIM23 does not affect PPARγ-mediated transcriptional activity in HEK293T cells. The peroxisome proliferator-activated receptor response element firefly luciferase reporter vector (*PPRE–Luc*), pGL4.74 renilla luciferase reporter plasmid, and expression vectors encoding *TRIM23* and *Pparg* were transfected into HEK293T cells and the cells were incubated in culture medium containing 10% charcoal-treated fetal bovine serum for 24 hr. Cells were incubated with or without troglitazone (2 μM) for 24 hr, harvested, and quantified firefly luciferase and renilla luciferase mRNA with real-time PCR. The data represent means ± s.d. from three independent experiments. (**B**) TRIM23 does not promote SUMOylation of PPARγ2. An in vivo assay for SUMOylation of PPARγ2 by TRIM23 was performed. Expression vectors encoding FLAG-PPARγ2, Myc-TRIM23, and HA-SUMO1 were transfected into HEK293T cells. Cell lysates were immunoprecipitated with anti-FLAG antibody and then immunoblot analysis was performed to detect modifications of PPARγ2.

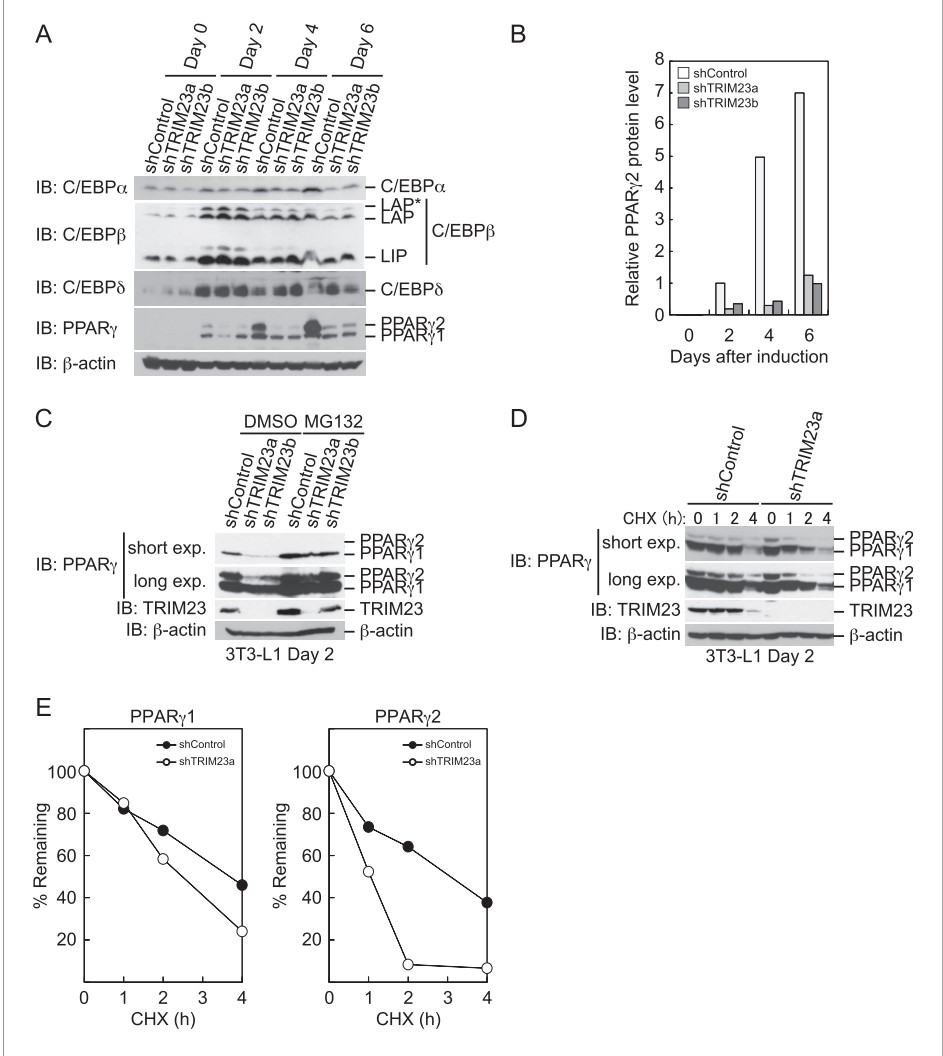

**Figure 7**. TRIM23 stabilizes PPARγ2. (**A**) Immunoblot analysis of C/EBPα, C/EBPβ, C/EBPδ and PPARγ proteins during 3T3-L1 cell differentiation. (**B**) The intensity of the immunoreactive bands of PPARγ2 obtained by immunoblot analysis with anti-PPARγ antibody was determined relative to that obtained with anti-β-actin antibody. (**C**) Immunoblot analysis of PPARγ protein in the absence and presence of MG132. 3T3-L1 cells with stable knockdown of TRIM23 or the corresponding control cells were differentiated by the differentiation cocktail for 48 hr and subsequently treated with 10 μM of MG132 for 6 hr (**D**) 3T3-L1 cells with stable knockdown of TRIM23 or the corresponding control cells were differentiated by the differentiation cocktail for 48 hr and subsequently treated with 10 μM of MG132 for 6 hr, followed by cycloheximide (CHX) treatment (5 μM) for 0, 1, 2 or 4 hr. The cell lysates were subjected to immunoblot analysis with an anti-PPARγ, anti-TRIM23 or anti-β-actin antibody. β-actin is shown as a loading control. The result is representative of two independent experiments. (**E**) The intensity of the PPARγ1 and PPARγ2 bands was normalized to that of the corresponding β-actin bands shown in (**D**) and is indicated as a percentage of the normalized value at 0 hr.

The following figure supplement is available for figure 7:

**Figure supplement 1**. The presence of TRIM23 is sufficient to inhibit basal degradation of PPARγ.

## TRIM23 functions as an E3 ubiquitin ligase for PPARγ2

Since it has been shown that TRIM23 is a putative E3 ubiquitin ligase and mediates atypical polyubiquitin conjugation to NEMO, we hypothesized that TRIM23 was involved in the stability of PPARγ protein via its E3 ligase activity (*Arimoto et al., 2010*). We next examined whether TRIM23

binds to and ubiquitinates PPARγ2. Using immunoprecipitation assays in HEK293T cells, we detected overexpressed HA-PPARγ2 in a protein complex with FLAG-TRIM23 (*Figure 8A*). This interaction was not affected by the treatment of troglitazone (*Figure 8B*). We also found that endogenous PPARγ2 was co-precipitated with FLAG-TRIM23 (*Figure 8C*). Furthermore, we found that TRIM23 facilitated ubiquitination of PPARγ2 in vivo and in vitro (*Figure 8D,E*). We next examined which domain of TRIM23 was required for adipocyte differentiation. 3T3-L1 cells were infected with a retrovirus expressing shRNA for TRIM23 (shTRIM23a) or a non-targeting control shRNA (shControl) and were subsequently infected with a retrovirus expressing control or shRNA-resistant FLAG-TRIM23 deletion mutants (*Figure 8F*). Expression levels of TRIM23 were verified by immunoblot analysis (*Figure 8G*). These cells were induced to differentiate, and the ability to undergo differentiation to mature adipocytes was evaluated by determination of lipid accumulation using Oil Red O staining. As expected, even a small amount of wild-type TRIM23 expression rescued lipid accumulation; however, a larger amount of TRIM23 deletion mutants failed to rescue lipid accumulation, indicating that both an amino-terminal RING finger domain, which is a ubiquitin ligase catalytic domain, and a carboxy-terminal ARF domain were required for adipocyte differentiation (*Figure 8H*). These results suggest that ubiquitin conjugation to PPARγ by TRIM23 plays an important role in adipocyte differentiation.

## TRIM23 mediates atypical polyubiquitin conjugation, leading to reduced recognition by the proteasomal ubiquitin receptor S5a

Degradative polyubiquitin chains are targeted to proteasomes through ubiquitin receptors including S5a/Rpn10 and Rad23. We next examined the binding of ubiquitinated PPARγ to GST-ubiquitin receptors and found that ubiquitinated PPARγ purified from cells exogenously expressing TRIM23 bound less efficiently to S5a than did that from cells without exogenous expression of TRIM23. In contrast, HR23B bound to ubiquitinated PPARγ from both cells with and those without exogenous expression of TRIM23 (*Figure 9A*). These findings indicate that TRIM23-dependent modification of PPARγ in vivo decreased its recognition by 26S proteasome in a manner dependent on the proteasome subunit S5a/Rpn10. It is well known that K48-linked ubiquitin chains are responsible for proteasomal degradation and that K63-linked chains are involved in various cell signaling processes, though the roles of other atypical ubiquitin linkages through M1, K6, K11, K27, K29 or K33 or mixed linkages within the same chain remain unclear (*Kulathu and Komander, 2012*). We hypothesized that TRIM23 mediates atypical ubiquitination of PPARγ to inhibit its degradation. We performed an in vitro ubiquitination assay using methylated ubiquitin or various ubiquitin mutants (all Lys mutated to Arg except the indicated Lys residue) to study the TRIM23-mediated ubiquitin-linkages of PPARγ. Intriguingly, TRIM23 mediated ubiquitin conjugation to PPARγ in the presence of methylated ubiquitin and amino-terminally His$_6$-tagged no Lys (K0) ubiquitin (*Figure 9B*). Considering that methylated ubiquitin is unable to form polyubiquitin chains and that amino-terminally His$_6$-ubiquitin mutants interfere with linear ubiquitin conjugation, these findings indicated that TRIM23 ubiquitinated PPARγ at multiple sites. Furthermore, TRIM23 more efficiently mediated ubiquitin conjugation in the presence of untagged K0 ubiquitin or His$_6$-K27 ubiquitin (*Figure 9B*), suggesting that TRIM23 can conjugate linear and K27-linked polyubiquitin chains to PPARγ. To confirm that this atypical ubiquitination of PPARγ is responsible for reduced recognition by 26S proteasome, we performed an in vitro binding assay using PPARγ ubiquitinated by TRIM23 and GST-ubiquitin receptors including HR23B and S5a. Consistent with the results shown in *Figure 9A*, although GST-HR23B efficiently bound to M1- and K27-Ub conjugates on PPARγ2, GST-S5a poorly pulled down the conjugates (*Figure 9C,D*). Taken together, these findings suggested that atypically ubiquitinated PPARγ by TRIM23 decreased its recognition by 26S proteasome, leading to resistance to proteasomal degradation.

## Ectopic expression of PPARγ2 rescues the adipogenic defect in TRIM23-knockdown 3T3-L1 cells

Our data suggest that the defect in adipogenesis by TRIM23 knockdown was caused at least through the decreased expression of PPARγ2. Therefore, we expected that overexpression of PPARγ2 at an abundance that is sufficient to overcome the instability of PPARγ2 protein should rescue the adipogenic defect by TRIM23 knockdown. To confirm this hypothesis, 3T3-L1 cells were infected with a retrovirus expressing shRNA for TRIM23 (shTRIM23a) or a non-targeting control shRNA (shControl) and were subsequently infected with a retrovirus expressing HA-PPARγ2 or control. Expression levels

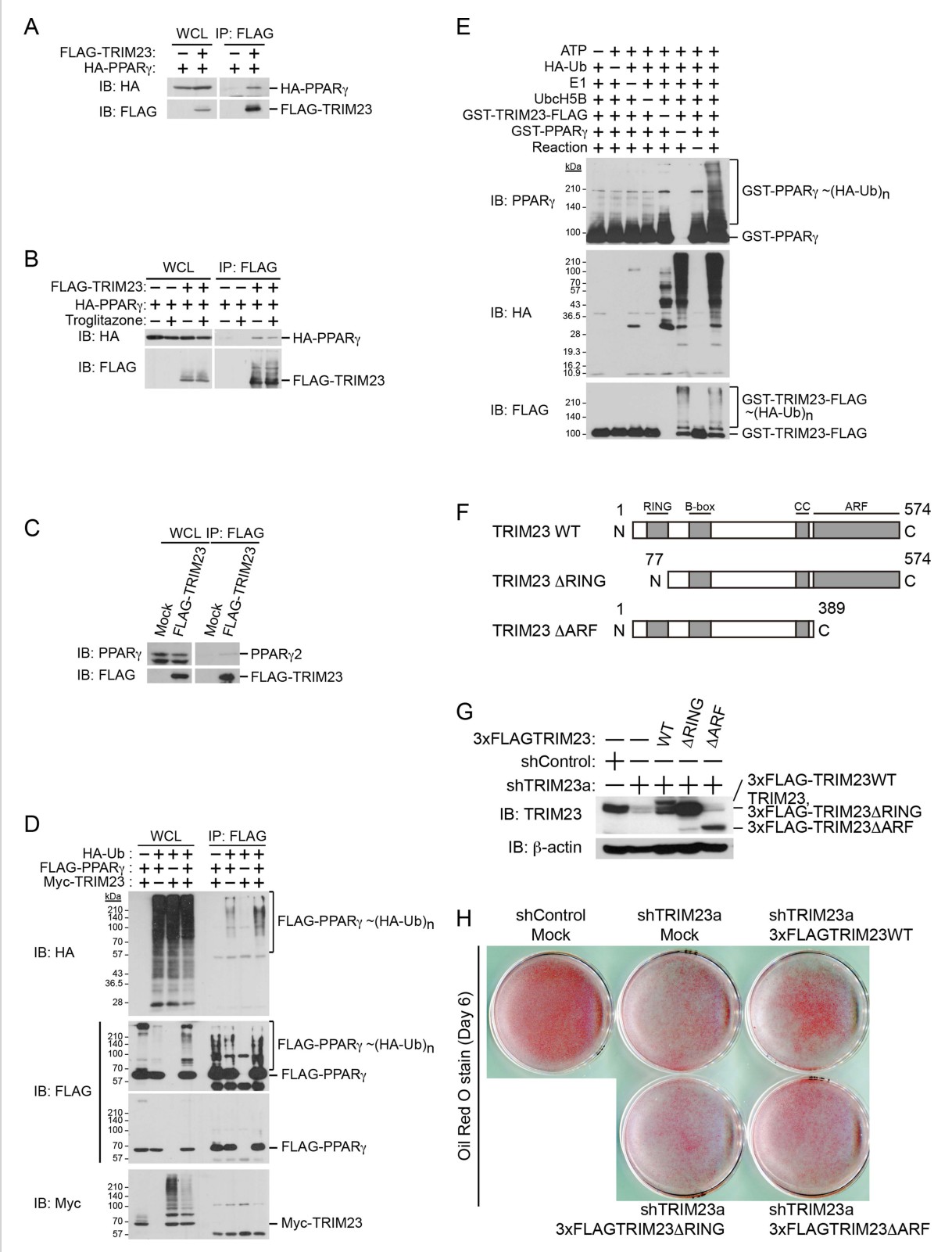

**Figure 8**. TRIM23 interacts with PPARγ2 and promotes ubiquitination of PPARγ2. (**A**) In vivo assay for interaction between TRIM23 and PPARγ2. FLAG-TRIM23 and HA-PPARγ were transfected into HEK293T cells. Lysates were immunoprecipitated with anti-FLAG antibody and immunoblotted with anti-FLAG and anti-HA antibodies. (**B**) In vivo assay for interaction between TRIM23 and PPARγ2 with or without troglitazone (1 μM). FLAG-TRIM23 and

*Figure 8. continued on next page*

*Figure 8. Continued*

HA-PPARγ2 were transfected into HEK293T cells. Coimmunoprecipitation assays were performed using the cell extract from these cells with anti-FLAG antibody in the presence or absence of 1 μM troglitazone. (**C**) In vivo assay for interaction between TRIM23 and PPARγ2. 3T3-L1 cells stably expressing FLAG-TRIM23 were generated, differentiated, and harvested at day 6. Whole cell lysates were immunoprecipitated with anti-FLAG antibody and then immunoblot analysis was performed with anti-FLAG and anti-PPARγ antibodies. (**D**) In vivo assay for ubiquitination of PPARγ2 by TRIM23. FLAG-PPARγ2, Myc-TRIM23, and HA-ubiquitin (HA-Ub) were transfected into HEK293T cells. Cell lysates were immunoprecipitated with anti-FLAG antibody and then immunoblot analysis was performed to detect ubiquitination of PPARγ2. (**E**) Promotion of in vitro PPARγ2 polyubiquitination by TRIM23. An in vitro ubiquitination assay was performed with the indicated combinations of ATP, HA-Ub, His$_6$-E1, His$_6$-E2 (UbcH5B), His$_6$-GST-TRIM23-FLAG, and GST-PPARγ2. Reaction mixtures were subjected to immunoblot analysis with anti-PPARγ (top), anti-HA (middle) or anti-FLAG (bottom). The positions of GST-PPARγ2 or His$_6$-GST-TRIM23-FLAG modified by various numbers of HA-Ub moieties are indicated. (**F**) Schematic representation of TRIM23 deletion mutants is shown. Protein motifs are indicated. RING, ring-finger domain; B-box, B-box domain; CC, coiled-coil domain; ARF, ADP ribosylation factor domain. (**G**) Immunoblot analysis of ectopic expression of TRIM23 deletion mutants in TRIM23-knockdown 3T3-L1 cells before induction of adipogenesis. (**H**) Cells were stained with Oil Red O to visualize the accumulation of lipid droplets at day 6.

of PPARγ2 and TRIM23 were verified by immunoblot analysis (*Figure 10A* and *Figure 10—figure supplement 1*). TRIM23 knockdown was maintained in spite of ectopic expression of PPARγ2 during adipocyte differentiation (*Figure 10B*). These cells were induced to differentiate, and the ability to undergo differentiation to mature adipocytes was evaluated by the determination of lipid accumulation using Oil Red O staining (*Figure 10C*) and determination of relative mRNA levels of adipocyte-specific genes (*Figures 10D* and *Figure 10—figure supplement 1*). Ectopic expression of PPARγ2 remarkably rescued lipid accumulation in TRIM23-knockdown cells and the induction of adipocyte-specific genes. The induction of early transcriptional factors, C/EBPβ and C/EBPδ, in TRIM23-depleted cells with ectopic expression of PPARγ2 was largely rescued to levels comparable to those in control 3T3-L1 cells during adipocyte differentiation, except for sustained high expression level of C/EBPβ after day 4 (*Figure 10—figure supplement 1*). These findings suggest that TRIM23 functions in the upstream stage of PPARγ induction during adipogenesis.

## Discussion

Adipocyte differentiation is tightly regulated by a complicated transcriptional cascade. PPARγ plays a central role in the late phase of adipogenesis. Thus, elucidation of the mechanisms that regulate PPARγ expression is essential for understanding adipocyte differentiation. In this study, we showed that TRIM23 knockdown in preadipocytes results in decreased expression of PPARγ and a severe defect in late adipogenic differentiation. Ectopic expression of PPARγ2 rescues the adipogenic defect induced by TRIM23 knockdown. Therefore, we further investigated the mechanism by which TRIM23 regulates the expression of PPARγ.

A considerable number of studies have shown the importance of transcriptional regulation of *Pparg* gene expression. Early adipogenic activators such as C/EBPβ and C/EBPδ provoked by adipogenic stimuli induce low expression levels of PPARγ1, PPARγ2 and C/EBPα (*Yeh et al., 1995*; *Ishibashi et al., 2012*). Then PPARγ and C/EBPα directly induce each other's high expression level that supports the maintenance of the differentiated stage (*Rosen et al., 2002*). The binding of activators to enhancer DNA elements promotes the recruitment of GTFs and Pol II to the promoter DNA in order to form a PIC (*Lemon and Tjian, 2000*; *Thomas and Chiang, 2006*). Activators also mediate the recruitment of covalent histone-modifying enzymes and ATP-driven nucleosome remodeling complexes to the promoter to reorganize the chromatin architecture to be competent for transcription, leading to transcription initiation (*Belotserkovskaya and Berger, 1999*; *Jaenisch and Bird, 2003*; *Margueron et al., 2005*). To elucidate the mechanism of reduced PPARγ expression by TRIM23 knockdown, we investigated the occupancy of activators such as C/EBPβ and C/EBPδ, epigenetic marks by ChIP-qPCR analysis and the degree of chromatin opening by FAIRE analysis at the *Pparg* promoters. However, there were no substantial changes in these during the early phase of differentiation, and we therefore concluded that recruitment of early activators and PIC formation were not affected by TRIM23. An apparent change observed in TRIM23-knockdown cells was a decrease in the occupancy of PPARγ and Pol II at the *Pparg* locus during the late phase of differentiation, during which PPARγ and C/EBPα support the high expression level of each other through a positive feedback loop. Therefore, this change caused by TRIM23 knockdown was likely to be provoked by impaired positive-feedback regulation of late adipogenic activators.

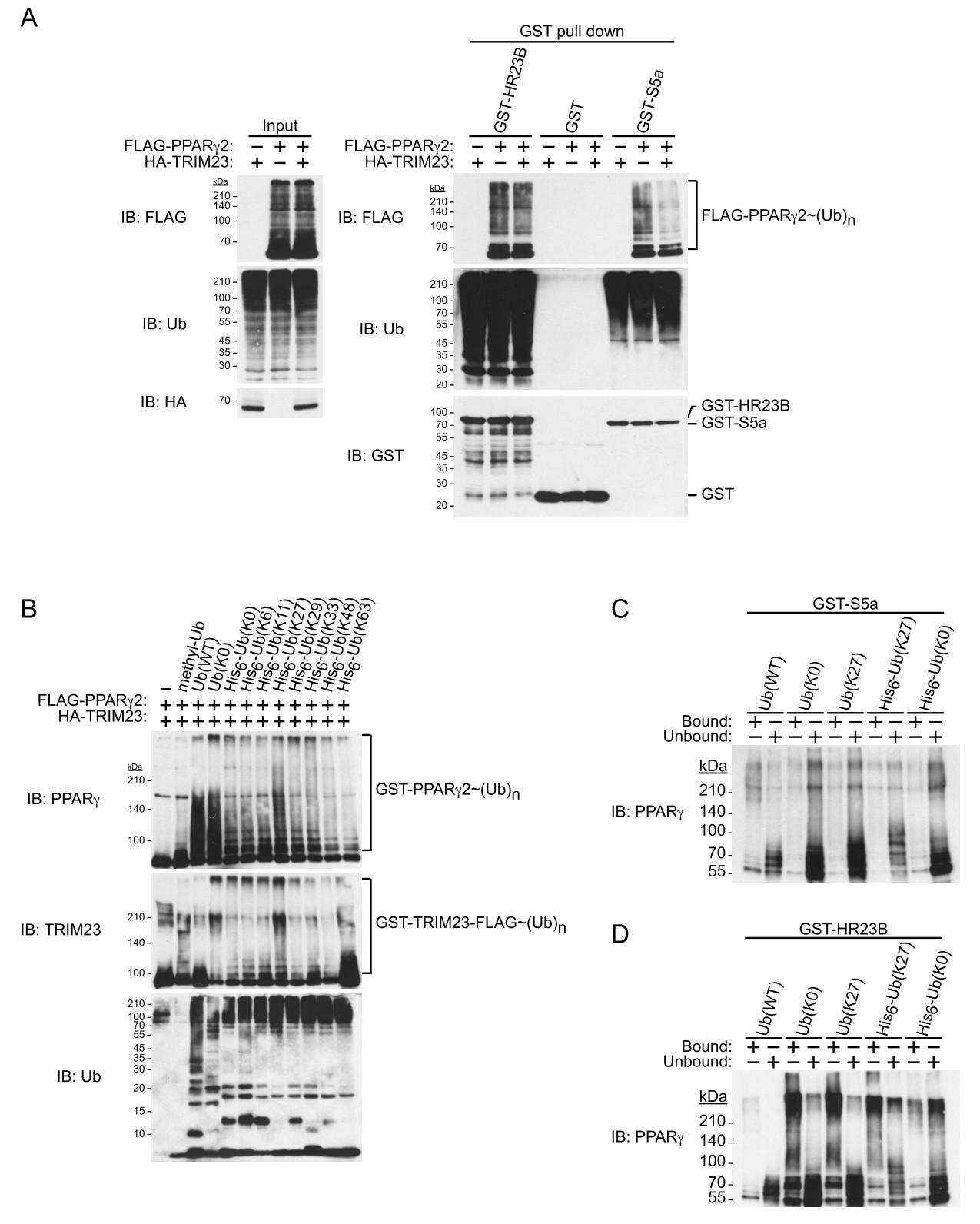

**Figure 9**. TRIM23 mediates atypical polyubiquitin conjugation of PPARγ2, leading to reduced recognition of ubiquitinated PPARγ2 by the proteasomal ubiquitin receptor S5a. (**A**) PPARγ2 polyubiquitination by TRIM23 leads to reduced recognition by the proteasomal ubiquitin receptor S5a. HEK293T cells transiently transfected with plasmids encoding FLAG-PPARγ2 and/or HA-TRIM23 were lysed. GST, GST-HR23B, and GST-S5a were resuspended with cell
*Figure 9. continued on next page*

*Figure 9. Continued*

lysates, followed by pull-down with glutathione Sepharose beads. Samples were separated by SDS-PAGE, followed by immunoblotting using the indicated antibodies. (**B**) In vitro PPARγ2 polyubiquitination by TRIM23. An in vitro ubiquitination assay was performed with the indicated combinations of ATP, various ubiquitin mutants, His$_6$-E1, E2 (UbcH5C), His$_6$-GST-TRIM23-FLAG, and GST-PPARγ2. Reaction mixtures were subjected to immunoblot analysis with anti-PPARγ (top), anti-TRIM23 (middle) or anti-Ub (bottom) antibodies. The positions of GST-PPARγ2 or His$_6$-GST-TRIM23-FLAG modified by various numbers of ubiquitin moieties are indicated. (**C** and **D**) Conjugation of PPARγ2 with M1- and/or K27-linked polyubiquitin chains leads to reduced recognition by the proteasomal ubiquitin receptor S5a. An in vitro ubiquitination assay was performed with the indicated combinations of ATP, ubiquitin mutants, His$_6$-E1, E2 (UbcH5C), TRIM23-FLAG, and PPARγ2. Reaction mixtures were subjected to GST pull-down assay. PPARγ2-ubiquitin conjugates were incubated with GST-S5a (**C**) or GST-HR23B (**D**) prebound to glutathione Sepharose beads. Samples were separated by SDS-PAGE, followed by immunoblotting using a PPARγ antibody. Equivalent amounts of bound and unbound fractions were loaded in each lane.

We examined the expression of adipogenic activators in detail, and we found that the protein level of PPARγ2 was already decreased at day 2 by TRIM23 knockdown (*Figure 7B*), whereas the mRNA level of *Pparg* was moderately reduced in TRIM23-knockdown cells (*Figure 3*). We therefore speculated that TRIM23 regulates PPARγ expression at the protein level. Although numerous studies have focused on the regulation of *Pparg* expression at the transcriptional level, information on regulation of PPARγ at the protein level remains limited. The stability of PPARγ protein is regulated by the ubiquitin proteasome pathway, and PPARγ is actually an unstable protein (t½ = 2 hr) (*Waite et al., 2001*; *Christianson et al., 2008*). Seven-in-absentia homolog 2 (Siah2) facilitates ubiquitination and degradation of PPARγ in vivo, but it is unclear whether Siah2 directly ubiquitinates PPARγ in vitro (*Kilroy et al., 2012*). It has also been reported that PPARγ by itself acts as an E3 ubiquitin ligase and mediates K48-linked polyubiquitination and degradation of p65, suggesting that PPARγ as an E3 ligase is critical to terminate NFκB signaling (*Hou et al., 2012*). PPARγ forms polyubiquitin chains without a substrate using UBCH3 as E2 in vitro, while it has not been determined whether PPARγ is autoubiquitinated or not. In this study, we showed that TRIM23 conjugates atypical polyubiquitin chains including M1- and K27-linked ubiquitin chains to PPARγ, leading to reduced binding to the proteasomal ubiquitin receptor S5a. Functions regulated by K48- and K63-linked ubiquitin chains have been extensively studied. Past studies have established that K48-linked chains are responsible for proteasomal degradation and that K63-linked chains are involved in various cell signaling processes. Recently, roles of other atypical ubiquitin linkages through K6, K11, K27, K29, K33, M1, or mixed linkages within the same chain have been shown (*Kulathu and Komander, 2012*). It has been reported that E3 ubiquitin ligase Smad ubiquitination regulatory factor 1 (Smurf1) prevents orphan nuclear receptor Nur77 degradation through mediating its atypical ubiquitination via K6 and K27 linkage (*Lin et al., 2014*). It has also been shown that RING type E3 ligase SCF$^{\beta-TrCP}$ stabilizes the oncoprotein Myc by ubiquitination that requires K33, K48 and K63-linked ubiquitin chains, which antagonizes SCF$^{Fbw7}$-mediated K48-linked polyubiquitin chain formation (*Popov et al., 2010*). Moreover, Ring1B generates self-atypical mixed K6-, K27-, and K48-linked non-proteolytic polyubiquitin chains on itself, whereas E6AP leads to its degradation via ubiquitination of the same residues of Ring1B (*Ben-Saadon et al., 2006*). In this study, we showed that TRIM23 mediates atypical polyubiquitin conjugation including M1- and K27-linked ubiquitin chains to PPARγ and that ubiquitination of PPARγ by TRIM23 causes reduced recognition of PPARγ by 26S proteasome. Although the detailed mechanism remains to be elucidated, these findings suggest that M1- and/or K27-linked polyubiquitin chains to PPARγ lead to resistance to degradation via reduced recognition by 26S proteasome.

In conclusion, TRIM23 is a novel positive regulator of adipocyte maturation via control of switching from early to late adipogenic enhanceosomes through regulating the abundance of PPARγ (illustrated in *Figure 10E*). Results of further studies on TRIM23 may be useful for revealing the abnormalities in adipocyte differentiation and for providing a potential therapeutic target for obesity and diabetes mellitus.

## Materials and methods

### Mice

All experiments were performed according to the guidelines laid down by the Animal Welfare Committee of Hokkaido University and under institutional approval. For diet-induced obesity, male

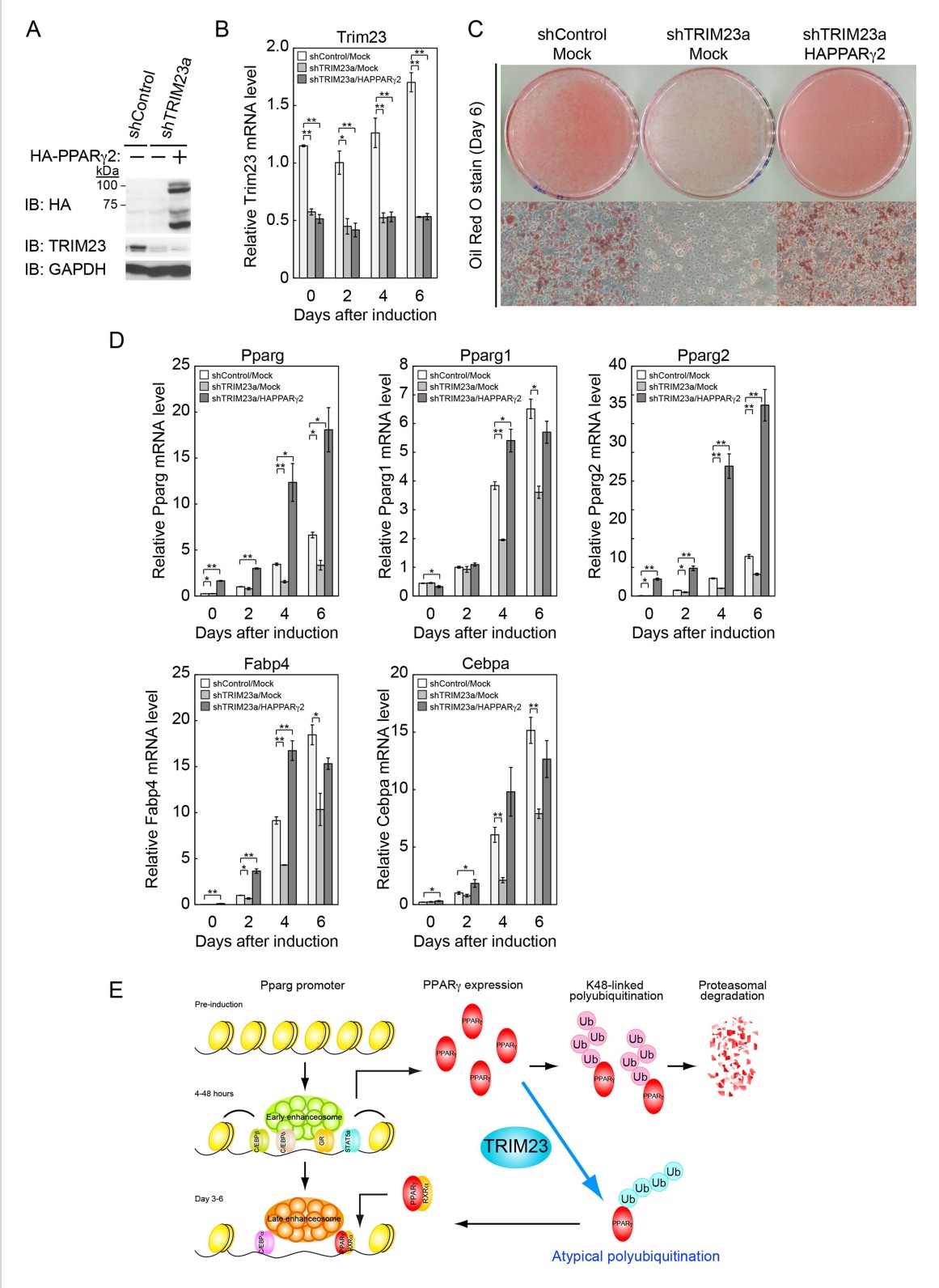

**Figure 10**. PPARγ2 expression rescues the adipogenesis defect in TRIM23-knockdown 3T3-L1 cells. (**A**) Immunoblot analysis of ectopic PPARγ2 expression before induction of adipogenesis is shown. (**B**) Changes in *Trim23* mRNA during adipocyte differentiation in 3T3-L1 cells. Total RNA was isolated from 3T3-L1 cells on the indicated days of differentiation. *Trim23* mRNA was determined by real-time PCR and normalized to that of *Gtf2b*. (**C**) Cells were stained with Oil Red O to visualize the accumulation of lipid droplets at day 6. (**D**) RNA levels of *Fabp4*, *Cebpa*, and *Pparg* were determined by real-time
*Figure 10. continued on next page*

Figure 10. Continued

PCR at days 0, 2, 4 and 6 of differentiation. Expression level of each gene was normalized to the level of *Gtf2b*. The data represent means ± s.d. from three independent experiments. The p values for the indicated comparisons were determined using Student's *t*-test (\*p < 0.05; \*\*p < 0.01). (**E**) Model for TRIM23 function in PPARγ abundance during adipocyte maturation. Several adipogenic stimuli activate early adipogenic activators such as C/EBPβ, C/EBPδ, glucocorticoid receptor (GR) and signal transducer and activator of transcription 5a (STAT5a). These activators then induce late adipogenic activators including PPARγ and C/EBPα. TRIM23 mediates atypical polyubiquitin conjugation to PPARγ and may stabilize the PPARγ protein. TRIM23 plays a critical role in the switching from early to late adipogenic enhanceosomes by regulating the abundance of PPARγ.

The following figure supplement is available for figure 10:

**Figure supplement 1**. PPARγ2 expression rescues the adipogenesis defect that occurred in TRIM23-knockdown 3T3-L1 cells.

C57BL/6 mice were fed high-fat diet containing a 57-kcal% fat high-fat diet (Clea Japan, Inc., Tokyo, Japan) for 27 weeks starting at 8 weeks of age (n = 3). Simultaneously, a separate group of male C57BL/6 mice were fed a diet of normal chow (n = 3) containing 5 kcal% fat and served as lean controls. The epididymal fat pads were dissected and minced and then placed in 1 mg/ml collagenase (Sigma–Aldrich, St. Louis, Missouri) in PBS and incubated at 37°C with shaking for 30 min. The samples were then centrifuged at 300×*g* for 5 min at room temperature. The floating fraction was collected as adipocytes, whereas the precipitated fraction was collected as the SVF.

## Cell culture and staining

HEK293T cells were cultured under an atmosphere of 5% $CO_2$ at 37°C in Dulbecco's modified Eagle's medium (Sigma–Aldrich) supplemented with 10% fetal bovine serum (FBS) (Gibco BRL, Paisley, UK). 3T3-L1 cells were cultured under the same conditions in DMEM with 10% calf serum (CS) (Equitech-Bio Inc., Kerrville, TX). Differentiation into mature adipocytes was induced by exposing the confluent cells to DMEM with 10% FBS for 2 days and to a conditioned medium supplemented with 0.5 mM IBMX, 1 µM dexamethasone, and 10 µg/ml insulin (induction medium) for 2 days. Cells were then incubated in a medium containing 5 µg/ml insulin. After 2 days, the medium was replaced with a conditioned medium. Subsequently, the conditioned medium was changed regularly every 2 days. Intracellular lipid accumulation was evaluated using Oil Red O staining. 3T3-L1 cells cultured on six-well plates were washed with PBS, fixed for 10 min with 10% formaldehyde in PBS, and stained for 30 min with Oil Red O solution (0.5% Oil Red O [Sigma–Aldrich] in isopropanol with Milli Q [3:2 vol/vol]). After the cells had been washed twice with 60% isopropanol and twice with PBS, lipid accumulation was evaluated. Intracellular triglyceride content was determined by Lab Assay Triglyceride (WAKO, Osaka, Japan) and normalized to the amounts of total cellular protein determined by a Bio-rad protein assay (Bio-Rad Laboratories Inc., Hercules, CA) according to each manufacturer's instructions. Culture and differentiation of human primary visceral preadipocytes (Poietics human visceral preadipocytes, Lonza Walkersville Inc., Walkersville, MD, USA) into adipocytes were performed according to the manufacturer's protocol.

## Cloning of cDNAs and plasmid construction

Human *TRIM23* and mouse *Pparg2* cDNAs were amplified by PCR from HEK293T and 3T3-L1 cDNAs, respectively, by polymerase chain reaction (PCR) with KOD plus (Toyobo, Osaka, Japan). The primers used for the amplification are listed in *Table 1*. TRIM23 and *Pparg2* cDNAs were ligated into the p3×FLAG vector (Invitrogen, Carlsbad, CA) and the pCGN-HA vector.

## Transfection, immunoprecipitation, and immunoblot analysis

HEK293T cells were transfected by the calcium phosphate method. After 48 hr, the cells were harvested and lysed in a solution containing 50 mM Tris-HCl (pH 7.4), 150 mM NaCl, 1% Triton X-100, leupeptin (10 µg/ml), 1 mM phenylmethylsulfonyl fluoride, 400 µM $Na_3VO_4$, 400 µM EDTA, 10 mM NaF, and 10 mM sodium pyrophosphate. The cell lysates were centrifuged at 16,000×*g* for 20 min at 4°C, and the resulting supernatant was incubated with antibodies for 2 hr at 4°C. Protein A-Sepharose (GE Healthcare, Uppsala, Sweden) that had been equilibrated with the same solution was added to the mixture, which was then rotated for 1 hr at 4°C. The resin was separated by centrifugation, washed 5 times with ice-cold lysis buffer, and then boiled in SDS sample buffer. Immunoblot analysis was

**Table 1.** List of SYBR Green primers for real-time PCR

| Gene | Forward primer | Reverse primer |
| --- | --- | --- |
| PCR primers for cloning of cDNA | | |
| Trim23 | AGGATGGCTACCCTGGTTGTAAAC | AAATCAAGCAACATCCAATACTCC |
| Pparg2 | GTTATGGGTGAAACTCTGGGA | CTGCTAATACAAGTCCTTGTA |
| Oligonucleotides for shRNA | | |
| shTRIM23a | GATCCCCGAAGAAATGGCTCTAAGTGTTCAAGAGACACTTAGAGCCATTTCTTCTTTTTA | AGCTTAAAAAGAAGAAATGGCTCTAAGTGTCTCTTGAACACTTAGAGCCATTTCTTCGGG |
| shTRIM23b | GATCCCCGGTAGATGTTAAATCGCATTTCAAGAGAATGCGATTTAACATCTACCTTTTTA | AGCTTAAAAAGGTAGATGTTAAATCGCATTCTCTTGAAATGCGATTTAACATCTACCGGG |
| qRT-PCR primers for gene expression | | |
| Adipoq | GCACTGGCAAGTTCTACTGCAA | GTAGGTGAAGAGAACGGCCTTGT |
| Cebpa | TGCGCAAGAGCCGAGATAA | CGGTCATTGTCACTGGTCAACT |
| Cebpb | CAAGCTGAGCGACGAGTACA | AGCTGCTCCACCTTCTTCTG |
| Cebpd | ATCGACTTCAGCGCCTACAT | GCTTTGTGGTTGCTGTTGAA |
| Cidec | AGCTAGCCCTTTCCCAGAAG | TCAGGCAGCCAATAAAGTCC |
| Fabp4 | CATCAGCGTAAATGGGGATT | GTCGTCTGCGGTGATTTCAT |
| Klf15 | CCCAATGCCGCCAAACCTAT | GAGGTGGCTGCTCTTGGTGTACATC |
| Pparg1 +2 | TGCAGGAGCAGAGCAAAGAG | CGGCTTCTACGGATCGAAAC |
| Pparg1 | TGAAAGAAGCGGTGAACCACTG | TGGCATCTCTGTGTCAACCATG |
| Pparg2 | TGGCATCTCTGTGTCAACCATG | GCATGGTGCCTTCGCTGA |
| Retn | TTTTCTTCCTTGTCCCTGAACTG | GATCTTCTTGTCGATGGCTTCAT |
| Gtf2b | GTTCTGCTCCAACTTTTGCCT | TGTGTAGCTGCCATCTGCACTT |
| Trim23 | TTGGAATGGCTCACACAGAAC | ACATGGGCATCAACAACAC |
| qPCR primers for ChIP | | |
| Pparg1 −1.0 kb | CTGTCTATCATGTGGGCTTCAG | ACCTTACACATAGGGTGGAGA |
| Pparg1 −0.4 kb | ACAAACTTCTCCATGACAGACA | CGCCTTGCTCCTCACAG |
| Pparg1 +0.3 kb | CTGCGTAACTGACAGCCTAAC | ACTTGGTCACTCTCCGTCCT |
| Pparg2 −1.0 kb | GATACACTGCCCTGTGTAAGG | GAGCAGCCCTTGTCACATAA |
| Pparg2 −0.2 kb | GAACAGTGAATGTGTGGGTCA | CTGACTGAGAGCCAGTTGTGA |
| Pparg2 +0.5 kb | GTGAGCATTTCAGAACACTTGG | GCCTGAAGAAGAACAGAAATTCTAC |
| Negative control | TGGTAGCCTCAGGAGCTTGC | ATCCAAGATGGGACCAAGCTG |
| Myog | GGGTCTCTTCCTCTTACCCGAT | ACCTTGCTGGCCATGGAC |
| Iqck | GAAACAAAGCCTTCCCATCC | TCCTTTCTTGCTGTGGCTTC |
| Cav2 | CTCAGAAAAGGCAGGGAAAG | CCCAGTCATGACAACACCAA |
| Fabp4 -10,000 bp | CCATGAGGAAATTCGCTACAC | CCTTCCACCCTTATCTCACAC |
| Fabp4 −5500 bp | GAGAGCAAATGGAGTTCCCAGA | TTGGGCTGTGACACTTCCAC |
| Fabp4 −200 bp | CATTGCCAGGGAGAACCAA | TCCTTCATGACCAGACCCTGT |
| Fabp4 +500 bp | CAGGTGAACCCGCAAGAAAG | GCTTGGCAAAGAAGGCCAC |

performed with primary antibodies, horseradish peroxidase-conjugated antibodies to mouse or rabbit immunoglobulin G (1:20,000 dilutions, Promega Corporation, Madison, WI) and an enhanced chemiluminescence system (ECL, Thermo Fisher Scientific, Rockford, IL). The following primary antibodies were used: anti-FLAG M2 (Sigma–Aldrich), anti-HA (Covance, Princeton, NJ), anti-PPARγ (sc-7273, Santa Cruz Biotechnology, Santa Cruz, CA), anti-TRIM23 (HPA039605, Sigma–Aldrich), anti-β-actin AC15 (Sigma–Aldrich), anti-GAPDH (Ambion, Austin, TX), anti-HDAC1 (10E2, Cell Signaling

Technology, Beverly, MA), anti-GST (sc-138, Santa Cruz), anti-Ub (P4D1, sc-8017, Santa Cruz), anti-C/EBPα (sc-61, Santa Cruz), anti-C/EBPβ (sc-150, Santa Cruz) and anti-C/EBPδ (sc-151, Santa Cruz). Nuclear extracts and cytoplasmic S100 fraction were prepared from 3T3-L1 cells according to the method of Dignam et al. (1983).

## Retrovirus expression system

A complementary DNA encoding mouse *Pparg2* containing an HA-tag at its amino terminus was subcloned into pMX-neo, and cDNAs encoding shRNA-resistant human TRIM23 deletion mutants containing a 3×FLAG-tag at their amino-terminus were subcloned into pMSCV-neo (Takara). The resulting vectors were used to transfect Plat-E cells and thereby generate recombinant retroviruses. 3T3-L1 cells were infected with the recombinant retroviruses and selected in a medium containing G418 (0.4 mg/ml, Nacalai Tesque, Kyoto, Japan).

## Measurement of PPARγ transcriptional activity using dual-luciferase

Cells were seeded in 24-well plates at $5 \times 10^4$ cells per well (HEK293T) and then incubated at 37°C with 5% $CO_2$ overnight. The peroxisome proliferator-activated receptor response element firefly luciferase reporter plasmid (PPRE–Luc, a kind gift from Bruce M Spiegelman) and pGL4.74 renilla luciferase (Promega) reporter plasmid were transfected with *Pparg* and/or *TRIM23* expression vectors into HEK293T cells using Fugene HD reagent (Roche, Branchburg, NJ). Transfected cells were incubated in DMEM (Invitrogen) supplemented with 10% charcoal-treated FBS (Equitech-Bio) for 24 hr and then incubated with 2 μM troglitazone for 24 hr and harvested, and firefly luciferase and renilla luciferase mRNA were quantified by real-time PCR.

## RNA interference

pSUPER-retro-puro vector was purchased from OligoEngine (OligoEngine, Seattle, WA). The sequences of short hairpin RNA (shRNA) oligonucleotides are listed in *Table 1* shRNA for a negative control with no significant homology to any known gene sequences in human and mouse genomes was designed according to a previous report (*Gou et al., 2004*). Approximately 50% confluent Plat-E cells in 100-mm dishes were transfected with 10 μg pSUPER-retro-puro-shTRIM23-a, shTRIM23-b or control shRNA vector using Fugene HD reagent (Roche). Culture supernatant containing the retrovirus was collected 48 hr after transfection, and retroviral supernatant was added to 3T3-L1 cell lines in 100-mm dishes with polybrene (8 μg/ml; Sigma–Aldrich). Cells were cultured with puromycin (5 μg/ml) for 1 week. Human primary visceral preadipocytes in six-well tissue culture plates ($\sim3.3 \times 10^5$ cells per well) were transfected with 12 nM siRNA targeting human TRIM23 (On-TARGET plus SMART pool, L-006523-00-0005, Dharmacon/Thermo Scientific, Lafayette, CO, USA) or with 12 nM ON-TARGETplus Non-targeting Pool (D-001810-10-20, Dharmacon) using Lipofectamine RNAiMAX Transfection Reagent (Invitrogen).

## Cell proliferation assay

Total cell number per dish was counted during a period of 6 days. 3T3-L1 cells were harvested from the 10-cm dish using 0.25% trypsin (Sigma–Aldrich) and 0.1 mM EDTA, and they were resuspended in a condition medium, followed by staining with 0.4% trypan blue (Sigma–Aldrich). Unstained cells were counted under a light microscope.

## RNA isolation and real-time PCR

Total RNA was isolated from 3T3-L1 cells using an ISOGEN (Nippon Gene, Tokyo, Japan), with removal of contaminating DNA by a TURBO DNA-Free Kit (Invitrogen), and reverse transcription (RT) was performed by ReverTra Ace (Toyobo, Osaka, Japan). The resulting cDNA was subjected to real-time PCR with a StepOne machine and Power SYBR Green PCR master mix (Applied Biosystems, Foster City, CA). The primers used for the amplification are listed in *Table 1*. *Gtf2b* transcript was used as an internal control to normalize the mRNA levels of each gene.

## In vitro ubiquitination assay

Reaction mixtures (each 30 μl) each containing 40 mM HEPES-NaOH (pH 7.9), 60 mM potassium acetate, 1 mM ATP, 5 mM $MgCl_2$, 0.5 mM EDTA, 2 mM DTT, E1 (0.1 μg; Enzo Life Sciences, Farmingdale, New York), E2 (0.5 μg of $His_6$-UbcH5B or 0.5 μg of $His_6$-UbcH5C), ubiquitin (4 μg of HA-Ub or 10 μg of other Ub mutants), TRIM23 (1 μg of $His_6$-GST-TRIM23-FLAG or 0.1 μg of TRIM23-FLAG),

and PPARγ (1 µg of His$_6$-GST-PPARγ or 0.1 µg of PPARγ) were incubated for 2 hr at 30°C. The reaction was terminated by the addition of SDS sample buffer containing 4% 2-mercaptoethanol and heating at 95°C for 5 min. Immunoblot analysis was performed with mouse monoclonal antibodies to Anti-FLAG M2 (Sigma–Aldrich), anti-HA (Covance) and anti-PPARγ (sc-7273, Santa Cruz Biotechnology).

## GST pull-down assay

HEK293T cells transiently transfected with plasmids encoding FLAG-PPARγ2 and/or HA-TRIM23 were harvested and lysed. PPARγ2-ubiquitin conjugates were prepared by an in vitro ubiquitination assay. GST pull-down assays were performed according to the instructions of the manufacturer (Glutathione Sepharose 4B, GE Healthcare). Briefly, 5 µg of GST (UBPBio, Aurora, CO, USA), GST-S5a (UBPBio) or GST-HR23B (UBPBio) was bound to Glutathione Sepharose beads (20 µl for each sample) for 1 hr, and the beads were then washed 3 times and resuspended in ice-cold lysis buffer. HEK293T cell lysates or PPARγ2-ubiquitin conjugates were incubated with the beads prebound by GST-tagged proteins for 1 hr at 4°C on a rotating platform. The supernatant was collected as an unbound fraction, and nonspecific binding was removed by washing 5 times with ice-cold lysis buffer. The bound proteins were eluted by 10 mM reduced glutathione. Samples were separated by SDS-PAGE, followed by immunoblotting using indicated antibodies. Equivalent amounts of bound and unbound fractions were loaded in adjacent lanes.

## Chromatin immunoprecipitation

Cells (1 × 10$^6$) were cross-linked with 0.5 M DSG in PBS for 45 min and 1% formaldehyde in PBS for 10 min at room temperature. Cells were resuspended in a hypotonic buffer (10 mM HEPES-NaOH pH 7.9, 1.5 mM MgCl$_2$, 10 mM KCl, and 0.5 mM DTT) and were homogenized using a dounce homogenizer. Crude nuclei were lysed in lysis buffer (0.1% SDS, 1% Triton X-100, 10 mM EDTA, 150 mM NaCl, 50 mM Tris-HCl pH 8.0) and were sonicated with a Bioruptor Sonicator (Diagenode, Sparta, NJ) 30 times for 30 s each time at the maximum power setting to generate DNA fragments of ~150–400 bps. Sonicated chromatin was incubated for 3 hr at 4°C with 5–15 µg of normal IgG or specific antibodies. Antibodies used were as follows: normal rabbit IgG (sc-2027, Santa Cruz), normal mouse IgG (sc-2025, Santa Cruz), anti-Pol II total Rpb1 (F-12, sc-55492, Santa Cruz), anti-C/EBPβ (sc-150, Santa Cruz), anti-C/EBPδ (sc-636, Santa Cruz), anti-PPARγ (sc-1984, Santa Cruz), anti-MED1 (sc-5334, Santa Cruz), H3K4me3 (07-473, EMD Millipore, Billerica, MA, USA), H3K27ac (ab4729, Abcam, Cambridge, MA), H3K9me2 (ab1220, Abcam) and H4K20me1 (ab9051, Abcam). Then protein A agarose (GE Healthcare) preblocked with BSA and salmon sperm DNA (BioDynamics Laboratory, Tokyo, Japan) was added and incubated for 1 hr at 4°C. Beads were washed once with IP buffer (20 mM Tris-HCl pH 8.0, 150 mM NaCl, 2 mM EDTA, 1% Triton X-100), 2 times with high-salt buffer (20 mM Tris-HCl pH 8.0, 500 mM NaCl, 2 mM EDTA, 1% Triton X-100), 2 times with LiCl buffer (250 mM LiCl, 20 mM Tris-HCl pH 8.0, 1 mM EDTA, 1% Triton X-100, 0.1% NP40 and 0.5% NaDOC), and 2 times with TE buffer. Bound complexes were eluted from the beads with 100 mM NaHCO$_3$ and 1% SDS by incubating at 50°C for 30 min with occasional vortexing. Crosslinking was reversed by overnight incubation at 65°C. Immunoprecipitated DNA and input DNA were treated with RNase A and Proteinase K by incubation at 45°C. DNA was purified using a QIAquick PCR purification kit (28106, QIAGEN, Valencia, CA) or MinElute PCR purification kit (28006, QIAGEN). Immunoprecipitated and input material was analyzed by quantitative PCR. ChIP signal was normalized to total input.

## Formaldehyde-assisted isolation of regulatory elements (FAIRE)

3T3-L1 cells (1 × 10$^6$) were cross-linked, and crude nuclei were isolated and sonicated as described for the ChIP assay (*Simon et al., 2012*). Samples were centrifuged and DNA in the supernatants was isolated by three extractions with phenol/chloroform/isoamyl alcohol (25:24:1). After the final extraction, the FAIRE samples were reverse cross-linked as described for the ChIP assay. The enrichment of fragmented genomic DNA in the FAIRE samples and input material was analyzed by quantitative PCR. FAIRE signal was normalized to total input.

## Statistical analysis

We used the unpaired Student's *t* test to determine the statistical significance of experimental data.

## Acknowledgements

We thank Susanne Mandrup for kindly providing a series of ChIP-qPCR primer sequences, Bruce M Spiegelman for *PPRE–Luc* reporter plasmid, Toshio Kitamura, Hisato Saito for plasmids and cell lines, Miho Uchiumi and Yuri Soida for help in preparing the manuscript.

## Additional information

### Funding

| Funder | Grant reference | Author |
|---|---|---|
| Japan Society for the Promotion of Science (JSPS) | KAKENHI, 24112006 | Shigetsugu Hatakeyama |
| Japan Society for the Promotion of Science (JSPS) | KAKENHI, 24390065 | Shigetsugu Hatakeyama |
| Japan Society for the Promotion of Science (JSPS) | KAKENHI, 10632424 | Masashi Watanabe |
| Japan Society for the Promotion of Science (JSPS) | KAKENHI, 25118501 | Hidehisa Takahashi |
| Suzuken Memorial Foundation | | Shigetsugu Hatakeyama |
| Naito Foundation | | Shigetsugu Hatakeyama |
| Uehara Memorial Foundation | | Shigetsugu Hatakeyama |

The funders had no role in study design, data collection and interpretation, or the decision to submit the work for publication.

### Author contributions

MW, Conception and design, Acquisition of data, Analysis and interpretation of data, Drafting or revising the article; HT, YS, SH, Conception and design, Analysis and interpretation of data, Drafting or revising the article; TO, SI, MS, WM, Acquisition of data, Analysis and interpretation of data; KT, Analysis and interpretation of data, Drafting or revising the article

### Ethics

Animal experimentation: All animal protocols were reviewed and approved by the Animal Welfare Committee of Hokkaido University. The work presented in this study is covered by the Animal Protocol Numbers APN-10-0077 and APN13-0040. All researchers, who performed procedures using live animal, were pre-approved by the Animal Welfare Committee of Hokkaido University, based on their completion of required animal use and care training, and acceptable previous experience in animal experiments.

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
