## [Decision Letter]

Thank you for sending your work entitled “TRIM23 regulates adipocyte differentiation via stabilizing PPARγ” for consideration at *eLife*. Your article has been favorably evaluated by Fiona Watt (Senior editor), a Reviewing editor, and three reviewers.

The Reviewing editor and the reviewers discussed their comments before we reached this decision, and the Reviewing editor has assembled the following comments to help you prepare a revised submission.

The finding that TRIM23 is a novel regulator of PPARγ, hypothetically through modulating polyubiquitin conjugation is of significant interest and if substantiated would have impact in this field and beyond. The experimental data presented provides a solid framework for how the hypothesis was derived, although Figure 2 alone solidly makes the case. However, the reviewers agree that additional key direct data is required to support the primary claim of the paper. The following two types of experiments are minimally necessary for publication:

1) What is the actual modification of PPARγ by TRIM23? What are the atypical ubiquitin chains that lead to the effect on PPARγ turnover?

2) Is ubiquitin conjugation to PPARγ actually mediating the effects observed? There is no direct evidence presented that ubiquitin conjugation to PPARγ by TRIM23 is the mechanism responsible for increased protein stability. There could be a different mechanism by which TRIM23 stabilizes PPARγ. Along these lines, the stabilization of PPARγ by TRIM23 in the K48R and K63R mutant background does not necessarily indicate that atypical ubiquitin conjugation is responsible for the stabilizing PPARγ.

A third point made by the reviewers that you should consider is the lack of data showing the effect of TRIM23 on PPARγ in vivo, although this point, unlike the above two issues, will not be necessary for publication.

---

## [Author Response]

1) What is the actual modification of PPARγ by TRIM23? What are the atypical ubiquitin chains that lead to the effect on PPARγ turnover?

We are grateful to the reviewer for raising this issue. It has been reported that TRIM23 is a putative E3 ubiquitin ligase (Arimoto, K.I. et al., 2010. Proc Natl Acad Sci USA. 107: 15856-15861) and that some TRIM family proteins have SUMO E3 ligase activities (Chu, Y. et al. 2011. Oncogene. 30:1108-1116). Since TRIM23 did not affect SUMOylation of PPARγ (Figure 6) and facilitated ubiquitination of PPARγ2 in vivo and in vitro (Figure 8), we focused on identification of the type of polyubiquitin chains of PPARγ by TRIM23. We performed an in vitro ubiquitination assay to clarify the type of polyubiquitin chain of PPARγ by TRIM23. In this assay, we took advantage of a series of ubiquitin (Ub) derivatives including wild-type Ub (Ub (WT)), methylated Ub (methyl-Ub), Ub (K0) in which all Lys are substituted with Arg, and His_6_-tagged ubiquitin mutants Ub (K0), Ub (K6), Ub (K11), Ub (K27), Ub (K29), Ub (K33), Ub (K48) and Ub (K63) in which the indicated Lys alone is not substituted with Arg (Figure 9). Since methylated ubiquitin is unable to form polyubiquitin chains and amino-terminally His_6_-tagging of ubiquitin interferes with linear ubiquitin conjugation, the assay using these mutated Ubs makes it possible to distinguish various types of ubiquitination including monoubiquitination, multi-ubiquitination and polyubiquitination with different chain types.

As shown in Figure 9, TRIM23 enhanced ubiquitin conjugation of PPARγ in the presence of Ub (WT), Ub (K0) and His_6_-tagged Ub (K27), indicating that M1 (linear)- and K27-linked polyubiquitin chains were formed by TRIM23. Of note, weak ubiquitination was observed with other mutants of ubiquitin including methyl-Ub, His_6_-tagged Ub (K0), Ub (K6), Ub (K11), Ub (K29), Ub (K33), Ub (K48) and Ub (K63), suggesting that multiple sites of PPARγ were conjugated with these ubiquitin mutants in this assay. Thus, these findings indicate that TRIM23 conjugates M1- and K27-linked polyubiquitin chains to PPARγ (in the subsection headed “TRIM23 mediates atypical polyubiquitin conjugation, leading to reduced recognition by the proteasomal ubiquitin receptor S5a”).

Next, we tested the possibility that these atypical ubiquitin chains of PRARγ by TRIM23 lead to the effect on PPARγ turnover. To address this question, we examined whether ubiquitin conjugation to PPARγ by TRIM23 leads to the acquisition of resistance to proteasomal degradation. Recruitment of ubiquitinated proteins to the 26S proteasome is a key step in proteasomal degradation. Several ubiquitin receptors that mediate this process have been identified. Of those, HR23B and S5a/Rpn10 have been shown to play a central role. We performed a GST pull-down assay using GST-tagged HR23B, GST-tagged S5a and HEK293T cell lysates expressing PPARγ and/or TRIM23 (Figure 9). Whereas GST-S5a efficiently pulled down ubiquitinated PPARγ in the absence of TRIM23, GST-S5a poorly pulled down ubiquitinated PPARγ in the presence of TRIM23. In contrast, GST-HR23B pulled down ubiquitinated PPARγ from both cells with and those without exogenous expression of TRIM23. These findings suggested that atypically ubiquitinated PPARγ by TRIM23 decreased its recognition by 26S proteasome in a manner dependent on the proteasome subunit S5a/Rpn10, leading to resistance to proteasomal degradation (in the subsection headed “TRIM23 mediates atypical polyubiquitin conjugation, leading to reduced recognition by the proteasomal ubiquitin receptor S5a”).

To verify that M1- and/or K27-linked ubiquitination of PPARγ is responsible for reduced recognition by 26S proteasome, we tested whether in vitro ubiquitination of PPARγ by TRIM23 affects its binding to GST-tagged ubiquitin receptors. Consistent with the results shown in Figure 9, although GST-HR23B efficiently bound to M1- and K27-Ub conjugates on PPARγ2, GST-S5a poorly pulled down the conjugates (Figure 9) (in the aforementioned subsection).

Collectively, the results show that TRIM23 conjugates M1- and K27-linked polyubiquitin chains to PPARγ in vitro and that PPARγ modified by TRIM23 has decreased recognition by 26S proteasome. Although the precise mechanism remains to be elucidated, these findings suggest that M1- and/or K27-linked polyubiquitin chains to PPARγ acquire resistance to degradation via reduced recognition by 26S proteasome.

*2) Is ubiquitin conjugation to PPARγ actually mediating the effects observed? There is no direct evidence presented that ubiquitin conjugation to PPARγ by TRIM23 is the mechanism responsible for increased protein stability. There could be a different mechanism by which TRIM23 stabilizes PPARγ. Along these lines, the stabilization of PPARγ by TRIM23 in the K48R and K63R mutant background does not necessarily indicate that atypical ubiquitin conjugation is responsible for the stabilizing PPARγ*.

We thank the reviewer for raising this issue. We agree with the reviewer and think that it is very important to show evidence that ubiquitin conjugation to PPARγ by TRIM23 is responsible for increased PPARγ stability. To directly address the question, it is important to clarify whether atypical ubiquitination-resistant PPARγ, in which ubiquitination sites by TRIM23 are mutated, is more unstable in cells. However, there is a technical difficulty in generating a PPARγ mutant, since it is difficult to distinguish the sites conjugated with degradative ubiquitin chains by other E3 ligases and with proteasome-resistant ubiquitin chains by TRIM23. Instead of performing the strategy, we tested the requirement of ubiquitin ligase activity of TRIM23 for adipocyte differentiation. As described in the revised manuscript (Figure 8), we added shRNA-resistant FLAG-TRIM23 deletion mutants back to TRIM23-knockdown 3T3-L1 cells and differentiated them to mature adipocytes. As expected, even a small amount of wild-type TRIM23 expression rescued lipid accumulation; however, a large amount of TRIM23 ΔRING mutant, in which E3 ubiquitin ligase activity is lacking, failed to rescue lipid accumulation (Figure 8). These findings support our idea that ubiquitin conjugation to PPARγ by TRIM23 plays an important role in adipocyte differentiation probably through increased PPARγ stability (in the subsection headed “TRIM23 functions as an E3 ubiquitin ligase for PPARγ2”).

To directly address whether atypical ubiquitin conjugation is responsible for the stability of PPARγ, we examined whether ubiquitin conjugation to PPARγ by TRIM23 leads to the acquisition of resistance to proteasomal degradation. As we stated in the response to #1, we tested whether ubiquitination of PPARγ by TRIM23 affects its binding ability to ubiquitin receptors including Rad23 and Rpn10/S5a. We performed a GST pull-down assay using GST-tagged HR23B, GST-tagged S5a and HEK293T cell lysates expressing PPARγ and/or TRIM23 (Figure 9). Whereas GST-S5a efficiently pulled down ubiquitinated PPARγ in the absence of TRIM23, GST-S5a poorly pulled down ubiquitinated PPARγ in the presence of TRIM23. In contrast, GST-HR23B equally pulled down ubiquitinated PPARγ regardless of TRIM23 coexpression. These results indicated that TRIM23-dependent modification of PPARγ in vivo decreased its recognition by 26S proteasome in a manner dependent on the proteasome subunit S5a/Rpn10 (in the subsection headed “TRIM23 mediates atypical polyubiquitin conjugation, leading to reduced recognition by the proteasomal ubiquitin receptor S5a”).

Furthermore, we performed a GST pull-down assay using PPARγ ubiquitinated by TRIM23 in vitro. Consistent with the results shown in Figure 9, although GST-HR23B efficiently bound to the PPARγ2 M1- and K27-Ub conjugates, GST-S5a poorly pulled down the conjugates (Figure 9). These results indicate that S5a has more stringent selectivity of ubiquitinated target proteins than does HR23B.

Taken together, the results show that the RING domain of TRIM23 was required for adipocyte maturation and that TRIM23-mediated atypical ubiquitinations decreased the recognition of PPARγ by 26S proteasome. These findings suggest that ubiquitination of PPARγ by TRIM23 causes the reduced recognition of PPARγ by 26S proteasome, leading to increased stability of PPARγ.

*A third point made by the reviewers that you should consider is the lack of data showing the effect of TRIM23 on PPARγ in vivo, although this point, unlike the above two issues, will not be necessary for publication*.

We agree with the reviewer and we think that it is important to elucidate the effect of TRIM23 on PPARγ in vivo. TRIM23 KO mice have been generated in Dr. Vaughan’s laboratory (Meza-Carmen, V. et al. 2011. Proc Natl Acad Sci USA. 108: 10454-10459). It has been reported that mice lacking the TRIM23/ARD1 gene appeared grossly normal and thus far have presented no obvious pathological phenotype or histopathology of major organs, including the brain, spleen, heart, kidneys, lung, and axial lymph nodes, or problems with breeding. Fatty liver without identified cause appeared in some older KO mice. Dysregulation of PPARγ is linked to severe insulin resistance, and reduced insulin sensitivity plays a major role in the pathogenesis of non-alcoholic fatty liver disease (NAFLD) (Janani, C. et al. 2015. Diabetes Metab Syndr. 9: 46-50; Than, N.N. et al. 2015. Atherosclerosis. 239: 192-202). Since we found that TRIM23 knockdown caused a marked decrease in PPARγ protein abundance in vitro, it is possible that dysregulation of PPARγ is one of the causes of fatty liver in TRIM23 KO mice.